# Accelerating Optimization via Differentiable Stopping Time

**Zhonglin Xie**
Beijing International Center for Mathematical Research
Peking University
zlxie@pku.edu.cn

**Yiman Fong**
Department of Industrial Engineering
Tsinghua University
fangym23@mails.tsinghua.edu.cn

**Haoran Yuan**
School of Mathematical Science
Peking University
yuanhr@stu.pku.edu.cn

**Zaiwen Wen**
Beijing International Center for Mathematical Research
Peking University
wenzw@pku.edu.cn

## Abstract

A common approach for accelerating optimization algorithms is to minimize the loss achieved in a fixed time, which enables a differentiable framework with respect to the algorithm's hyperparameters. In contrast, the complementary objective of minimizing the time to reach a target loss is traditionally considered non-differentiable. To address this limitation, we propose a differentiable discrete stopping time and theoretically justify it based on its connection to continuous differential equations. We design an efficient algorithm to compute its sensitivities, thereby enabling a new differentiable formulation for directly accelerating algorithms. We demonstrate its effectiveness in applications such as online hyperparameter tuning and learning to optimize. Our proposed methods show superior performance in comprehensive experiments across various problems, which confirms their effectiveness.

## 1 Introduction

Optimization algorithms are fundamental to a wide range of applications, including operations research [1], the training of large language models [2], and decision-making in financial markets [3]. Consequently, significant research effort has been dedicated to accelerating these algorithms. A common formulation for algorithm design and tuning, prevalent in areas like hyperparameter optimization [4] and learning to optimize (L2O) [5], is to minimize the objective function value achieved after a fixed number of iterations or a predetermined time budget. This approach often leads to differentiable training objectives with respect to algorithmic hyperparameters, enabling gradient-based optimization of the algorithm itself.

However, this formulation does not directly optimize the number of iterations required to reach a desired performance level or target loss, which is often the practical goal in deployment. This complementary objective, minimizing the time to reach a target loss, is traditionally perceived as non-differentiable with respect to algorithm parameters, as stopping time is typically an integer-valued

39th Conference on Neural Information Processing Systems (NeurIPS 2025).

or non-smooth function of parameters, addressed only conceptually or via zeroth-order optimization methods [6].

To overcome this fundamental challenge and enable the direct, gradient-based optimization of convergence speed towards a target accuracy for iterative algorithms, this paper introduces the concept of differentiable stopping time. We propose a comprehensive framework that allows for the computation of sensitivities of the number of iterations required to reach a stopping criterion with respect to algorithm parameters. Our main contributions are summarized as follows:

- We formulate a new class of differentiable objectives for algorithm acceleration, aiming to directly minimize the number of iterations or computational time required to achieve a target performance. This is supported by a theoretical framework that establishes the differentiability of discrete stopping time via a connection between discrete-time iterative algorithms and continuous-time dynamics, leveraging tools from the theory of continuous stopping times.

- We develop a memory-efficient and scalable algorithm for computing sensitivities of discrete stopping time, enabling effective backpropagation through iterative procedures. Our experimental results validate the accuracy and efficiency of the proposed method, particularly in high-dimensional settings, and show clear advantages over approaches relying on exact ordinary differential equation solvers.

- We demonstrate the applicability of differentiable stopping time in practical applications, including L2O and the online adaptation of optimizer hyperparameters. These case studies show that differentiable stopping time can be seamlessly integrated into existing frameworks, and our empirical evaluations suggest that it provides a principled and effective lens for understanding and improving algorithmic acceleration.

## 1.1 Related Work

**ODE Perspective of Accelerated Methods.** Offering a continuous-time view of optimization algorithms, this perspective provides both theoretical insights and practical improvements. The foundational work [7] established a connection between Nesterov's accelerated gradient method and a second-order ordinary differential equation, introducing a dynamical systems viewpoint for understanding acceleration. Building on this, acceleration phenomena have been analyzed through high-resolution differential equations [8], revealing deeper insights into optimization dynamics. The symplectic discretization of these high-resolution ODEs [9] has also been explored, leading to practical acceleration techniques with theoretical guarantees. A Lyapunov analysis of accelerated gradient methods was developed in [10], extending the framework to stochastic settings. For optimization on parametric manifolds, accelerated natural gradient descent methods have been formulated in [11], based on the ODE perspective.

**Implicit Differentiation in Deep Learning.** This technique enables efficient gradient computation through complex optimization procedures. A modular framework for implicit differentiation was presented in [12], unifying existing approaches and introducing new methods for optimization problems. In non-smooth settings, [13] developed a robust theory of nonsmooth implicit differentiation with applications to machine learning and optimization. Implicit differentiation has also been applied to train iterative refinement algorithms [14], treating object representations as fixed points. For non-smooth convex learning problems, fast hyperparameter selection methods have been developed using implicit differentiation [15]. Training techniques for implicit models that match or surpass traditional approaches have been explored in [16], leveraging implicit differentiation. Implicit bias in overparameterized bilevel optimization has been investigated in [17], providing insights into the behavior of implicit differentiation in high-dimensional settings. In optimal control, implicit differentiation for learning problems has been revisited in [18], where new methods for differentiating optimization-based controllers are proposed.

**Learning to Optimize.** This emerging paradigm leverages machine learning techniques to design optimization algorithms. A comprehensive overview of L2O methods [19] categorizes the landscape and establishes benchmarks for future research. The scalability of L2O to large-scale optimization problems has been explored in [20], showing that learned optimizers can effectively train large neural networks. To enhance the robustness of learned optimizers, policy imitation techniques were introduced in [21], significantly improving L2O model performance. Generalization capabilities

have been studied by developing provable bounds for unseen optimization tasks [22]. In [23], meta-learning approaches are proposed for fast self-adaptation of learned optimizers. The theoretical foundations of L2O have been strengthened through convergence guarantees for robust learned optimization algorithms [24]. Furthermore, L2O has been extended to the design of acceleration methods by leveraging an ODE perspective of optimization algorithms [25].

## 2 Differentiable Stopping Time: From Continuous to Discrete

We consider an iterative algorithm that arises from the discretization of an underlying continuous-time dynamical system. Let $\mathcal{A}(\theta, x, t)$ be a function that defines the instantaneous negative rate of change for the state $x \in \mathbb{R}^d$ at time $t$, parameterized by $\theta$. The input $\theta$ could represent, for example, parameters of a step size schedule or weights of a learnable optimizer. Given $t_0$ and $x_0 \in \mathbb{R}^d$, the continuous-time dynamics are given by the ordinary differential equation (ODE)

$$\dot{x}(t) = -\mathcal{A}(\theta, x(t), t), \quad \text{with initial condition } x(t_0) = x_0. \tag{1}$$

The trajectory $x(t)$ aims to minimize a function $f(x)$, and $\mathcal{A}$ is typically related to $f(x)$. Applying the forward Euler discretization method to the ODE (1) with a time step $h > 0$ yields the iterative algorithm

$$x_{k+1} = x_k - h\mathcal{A}(\theta, x_k, t_k), \tag{2}$$

where $x_k$ is the approximation of $x(t_k)$, and $t_k = t_0 + kh$. We emphasize that $h$ serves as the discretization step for the ODE. The "effective step size" of the optimization algorithm at iteration $k$ is $h$ times any scaling factors embedded within $\mathcal{A}(\theta, x_k, t_k)$. We provide two simple examples of (2) as follows. This model also captures more sophisticated algorithms, such as the gradient method with momentum and LSTM-based learnable optimizers, as illustrated in Appendix D.

Figure 1 provides a visual intuition for these concepts. Figure 1a illustrates how the hyperparameter $\theta$ influences the optimization path. The solid lines are the idealized continuous trajectories from the ODE, while the dotted lines with markers show the actual discrete steps of the algorithm. The level sets of the stopping criterion are shown as concentric ellipses. Changing $\theta$ from 0.5 (red) to 2.5 (green) alters the trajectory, changing where and when the path intersects the stopping criterion. Figure 1b demonstrates a central idea of our work: the stopping time ($T_J$ for continuous, $N_J$ for discrete) is a smooth, differentiable function of the hyperparameter $\theta$. The close alignment between the discrete stopping times ($N_J$, transparent markers) and their continuous counterparts ($T_J$, solid lines) visually validates our ODE-based approximation and shows the key property our method exploits.

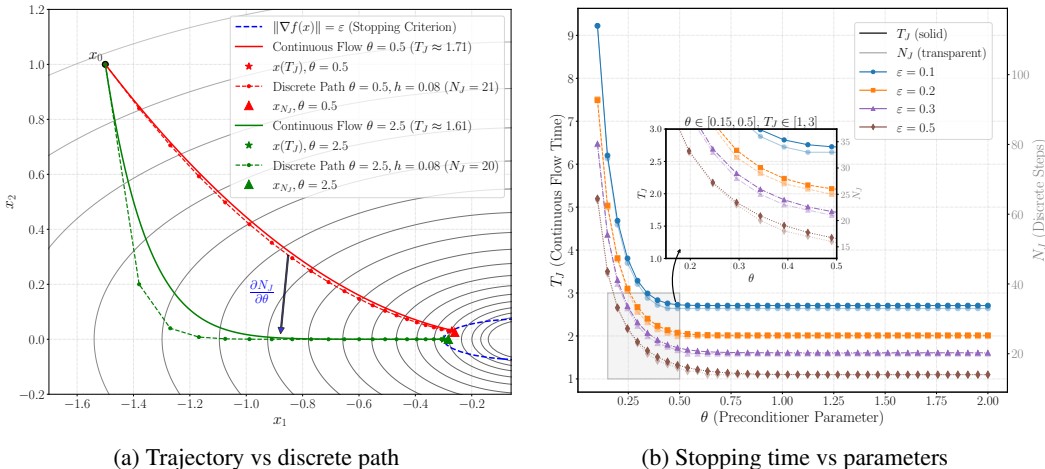

(a) Trajectory vs discrete path  (b) Stopping time vs parameters

Figure 1: Illustration of the differentiable stopping time on $f(x_1, x_2) = 0.5x_1^2 + 2x_2^2$ and $\mathcal{A}(x, \theta, t) = \text{diag}(1, \theta)\nabla f(x)$. Effect of $\theta$ on continuous and discrete stopping time $T_J, N_J$ for different $\varepsilon$ values.

**Rescaled Gradient Flow.** A common instance is the rescaled gradient flow, where $\mathcal{A}$ incorporates a time-dependent and parameter-dependent scaling factor $\alpha(\theta, t)$ for the gradient of the objective

function $f(x)$. In this case

$$\mathcal{A}(\theta, x(t), t) = \alpha(\theta, t)\nabla f(x(t)). \qquad (3)$$

The ODE becomes $\dot{x}(t) = -\alpha(\theta, t)\nabla f(x(t))$. The parameters $\theta$ might define the functional form of $\alpha$, e.g., if $\alpha(\theta, t) = \theta_1 e^{-\theta_2(t-t_0)}$, then $\theta = (\theta_1, \theta_2)$. The effective step size for the discretized iteration $x_{k+1} = x_k - h\alpha(\theta, t_k)\nabla f(x_k)$ is $h \cdot \alpha(\theta, t_k)$.

**Learned Optimizer.** Another relevant scenario involves the optimizer using a parametric model, such as a neural network. Let $\mathcal{N}(\cdot; \theta): \mathbb{R}^d \times \mathbb{R}^d \times \mathbb{R} \to \mathbb{R}^d$ denote a neural network parameterized by weights $\theta$. This network could learn, for instance, a diagonal preconditioning matrix $\mathrm{diag}(\mathcal{N}(x, \nabla f(x), t; \theta))$. Then, the function $\mathcal{A}$ is defined as

$$\mathcal{A}(\theta, x(t), t) = \mathrm{diag}(\mathcal{N}(x(t), \nabla f(x(t)), t; \theta))\nabla f(x(t)). \qquad (4)$$

The discretized update would then use this learned preconditioned gradient.

## 2.1 Differentiating the Continuous Stopping Time

We first give a formal definition of the continuous stopping time.

**Definition 1** (Continuous Stopping Time). *Given a continuously differentiable function J, for a stopping criterion defined by the condition $J(x) = \varepsilon$, the continuous stopping time is the first time that the trajectory reaches it:*

$$T_J(\theta, x_0, \varepsilon) := \inf_t \{t \mid J(x(t)) \leq \varepsilon, t \geq t_0, x(t) \text{ solves } (1)\}. \qquad (5)$$

*When $J(x(t))$ never reaches the target $\varepsilon$, we set $T_J(\theta, x_0, \varepsilon) = +\infty$.*

Now, we present a theorem that establishes conditions for the differentiability of the stopping time $T_J$ with respect to parameters $\theta$ or the initial condition $x_0$.

**Theorem 1** (Differentiability of Continuous Stopping Time). *Let $T = T_J(\theta, x_0, \varepsilon)$ be the continuous stopping time such that $J(x(T)) = \varepsilon$. We assume that the function $\mathcal{A}(\theta, x, t)$ is continuously differentiable with respect to $\theta$, $x$, and $t$. Additionally, the stopping criterion function $J(x)$ is assumed to be continuously differentiable with respect to $x$. Finally, it is assumed that the time derivative of the criterion function along the trajectory does not vanish at time $T$, that is,*

$$\frac{\mathrm{d}}{\mathrm{d}t}J(x(t))\Big|_{t=T} = \nabla J(x(T))^\top \dot{x}(T) \neq 0.$$

*Then, the solution $x(t; \theta, x_0)$ of the ODE system is continuously differentiable with respect to its arguments $\theta$ and $x_0$ for $t$ in a neighborhood of $T$. The stopping time $T_J(\theta, x_0, \varepsilon)$ is continuously differentiable with respect to $\theta$ (and $x_0$) in a neighborhood of the given $(\theta, x_0)$ where $T_J < \infty$. Specifically, its derivatives with respect to a component $\theta$ and $x_0$ are given by*

$$\frac{\partial T_J}{\partial \theta} = -\frac{\nabla J(x(T))^\top \partial x(T)/\partial \theta}{\nabla J(x(T))^\top \dot{x}(T)}, \qquad \frac{\partial T_J}{\partial x_0} = -\frac{\nabla J(x(T))^\top \partial x(T)/\partial x_0}{\nabla J(x(T))^\top \dot{x}(T)}. \qquad (6)$$

The differentiability of $x(T)$ with respect $\theta$ and $x_0$ is guaranteed by the smooth dependence of solutions on initial conditions and parameters. The terms $\partial x(T)/\partial \theta$ and $\partial x(T)/\partial x_0$ are sensitivities of the state $x$ at time $T$ with respect to $\theta$ and $x_0$ respectively, which can be obtained by solving the corresponding sensitivity equations or via adjoint methods. The proof is an application of the implicit function theorem, which is deferred to Appendix A. For the differentiability under weaker conditions, one may refer to [25, Proposition 2], which confirms the path differentiability [26] when $\mathcal{A}$ involves non-smooth components.

## 2.2 Differentiable Discrete Stopping Time: An Effective Approximation

The continuous stopping time $T_J$ is an ideal measure of algorithm efficiency. However, backpropagating through it requires solving forward and backward (adjoint) differential equations numerically. This can incur significant computational overhead, and many of the detailed steps evaluated by an ODE solver might be considered "wasted" compared to the coarser steps of the original iterative algorithm (2). We now return to the discrete iteration (2) and propose an efficient approach to approximate $\nabla_\theta T_J$ and $\nabla_{x_0} T_J$.

**Definition 2** (Discrete Stopping Time). *For a stopping criterion $J(x) = \varepsilon$ and the iterative algorithm (2), the discrete stopping time is the smallest integer such that $J(x_K) \leq \varepsilon$:*

$$N_J(\theta, x_0, \varepsilon) := \min_n \{n \mid J(x_n) \leq \varepsilon, n \geq 0, \{x_n\}_{n=0}^{\infty} \text{ satisfies (2)}\}. \tag{7}$$

*If $J(x_n) > \varepsilon$ for all $n \geq 0$, we set $N_J = +\infty$.*

While $N_J$ is inherently an integer, to enable its use in gradient-based optimization of $\theta$ or $x_0$, we seek a meaningful way to define its sensitivity to these parameters. Our approach is inspired by Theorem 1. We conceptualize $N$ as a continuous variable for which the condition $J(x_N(\theta, x_0)) \approx \varepsilon$ holds exactly at the stopping time. Formally differentiating this identity with respect to $\theta$ gives

$$0 \approx \nabla J(x_N)^\top \left( \frac{\partial x_N}{\partial N} \frac{\partial N}{\partial \theta} + \frac{\partial x_N}{\partial \theta} \right) \approx \frac{J(x_N) - J(x_{N-1})}{h} \frac{\partial N}{\partial \theta} + \nabla J(x_N)^\top \frac{\partial x_N}{\partial \theta}.$$

Under suitable regularity assumptions, this approximation allows us to define the sensitivity of the discrete stopping time.

**Definition 3** (Sensitivity of the Discrete Stopping Time). *Assume that the conditions of Theorem 1 hold. Let $N = N_J(\theta, x_0, \varepsilon)$ denote the discrete stopping time. Then, the sensitivities of $N$ with respect to $\theta$ and $x_0$ are defined as*

$$\frac{\partial N}{\partial \theta} := -\frac{h \nabla J(x_N)^\top \partial x_N / \partial \theta}{J(x_N) - J(x_{N-1})}, \qquad \frac{\partial N}{\partial x_0} := -\frac{h \nabla J(x_N)^\top \partial x_N / \partial x_0}{J(x_N) - J(x_{N-1})}. \tag{8}$$

Since Definition 2 ensures $J(x_N) - J(x_{N-1}) < 0$, the above expressions are well-defined. Beyond being a natural symbolic differentiation of the discrete stopping condition, this definition also serves as an effective approximation of the gradient of the continuous stopping time. The next theorem formalizes this connection by quantifying the approximation error between the sensitivities of the discrete stopping time and the gradients of the continuous stopping time.

**Theorem 2** (Approximation Error for Gradient of Stopping Time). *Let $J(x) = \varepsilon$ be a stopping criterion, and let $h > 0$ be a time step size. Assume the discrete stopping index $N_J$ satisfies $T_J \in (t_0 + (N_J - 1)h, \ t_0 + N_J h]$, where $T_J$ is the (continuous) stopping time and $t_0$ is the initial time. Suppose that the function $\mathcal{A}$ is twice continuously differentiable. We assume that $\mathcal{A}(\theta, x(t), t)$, regarded as a function of $(\theta, t)$, has uniformly bounded $W^{2,\infty}$ (Sobolev) norms with respect to $(\theta, t)$ in a neighborhood of $\theta \times [t_0, \ t_0 + N_J h]$, and that $J$, regarded as a function of $x$, has a uniformly bounded $W^{2,\infty}$ norm in a neighborhood of $x(T_J)$. Furthermore, suppose the boundary condition $\nabla J(x(T_J))^\top \dot{x}(T_J) \neq 0$ holds. Then, for sufficiently small $h$, the following holds*

$$\|\nabla_\theta T_J(\theta, x_0, \varepsilon) - \nabla_\theta N_J(\theta, x_0, \varepsilon)\| = \mathcal{O}(h). \tag{9}$$

This theorem demonstrates that Definition 3 provides an approximation to the gradient of the continuous stopping time. That is, $\nabla_\theta N_J$ converges to $\nabla_\theta T_J$ as $h \to 0$. An analogous result holds for the gradient with respect to $x_0$. This result serves as a theoretical justification for using the symbolic discrete sensitivity (8) as a surrogate for the continuous counterpart. Our $\mathcal{O}(h)$ error bound in (9) relies on the standard local error analysis of the Euler method. However, we note a connection to a non-trivial result from [25], which proves for certain forms of $\mathcal{A}$ that the *global error* $\|x_k - x(t_k)\|$ can converge to zero as $k \to \infty$ even with a *fixed, non-vanishing* step size $h$. While proving this for our more general framework is beyond the current scope, it suggests that the approximation in (9) may be more accurate than the local analysis implies, paving the way for future work to establish stronger error bounds that do not require $h \to 0$.

## 2.3 Efficient Computation of the Sensitivity

The primary challenge in (8) is computing the numerator term $\nabla J(x_N)^\top \partial x_N / \partial \theta$. If the function $\mathcal{A}$ and the iteration process are implemented within an automatic differentiation framework (e.g., PyTorch, TensorFlow) where $\mathcal{A}$ might be a learnable `nn.Module`, then the numerator can be obtained by unrolling the computation graph and applying backpropagation. However, this can be unstable and memory-intensive for large $N$. Other methods include finite differences or stochastic gradient estimators, which are inexact.

Alternatively, the discrete adjoint method provides a memory-efficient way to compute the required vector-Jacobian products $\nabla J(x_N)^\top (\partial x_N/\partial \theta)$ and $\nabla J(x_N)^\top (\partial x_N/\partial x_0)$ without forming the Jacobians explicitly. This method involves a forward pass to compute the trajectory $x_0, \ldots, x_N$, followed by a backward pass that propagates adjoint (or co-state) vectors. Let $x_{k+1} = G_k(x_k, \theta) = x_k - h\mathcal{A}(\theta, x_k, t_k)$ be the iterative update. Algorithm 1 outlines the procedure to compute the term $S_\theta = \nabla J(x_N)^\top (\partial x_N/\partial \theta)$ and $S_{x_0} = \nabla J(x_N)^\top (\partial x_N/\partial x_0)$. The correctness of Algorithm 1 is established by the following theorem. The proof is presented in Appendix C.

---

**Algorithm 1** Discrete Adjoint Method for Sensitivity Components

---

1: **Input:** Forward trajectory $\{x_k\}_{k=0}^N$, parameters $\theta$, $J(x_N)$, time step $h$, initial time $t_0$.
2: **Output:** $S_\theta = \nabla J(x_N)^\top (\partial x_N/\partial \theta)$ and $S_{x_0} = \nabla J(x_N)^\top (\partial x_N/\partial x_0)$.
3: $\lambda \leftarrow \nabla J(x_N)$.          ▷ Initialize adjoint vector
4: $S_\theta \leftarrow \mathbf{0}$ (vector of same size as $\theta$).      ▷ Initialize sensitivity component for $\theta$
5: **for** $k = N - 1$ **downto** 0 **do**
6:      $t_k \leftarrow t_0 + kh$.
7:      $S_\theta \leftarrow S_\theta - h\left(\frac{\partial \mathcal{A}(\theta, x_k, t_k)}{\partial \theta}\right)^\top \lambda$.         ▷ Accumulate contribution to $S_\theta$
8:      $\lambda \leftarrow \left(I - h\frac{\partial \mathcal{A}(\theta, x_k, t_k)}{\partial x_k}\right)^\top \lambda$.         ▷ Propagate adjoint vector backward
9: **end for**
10: $S_{x_0} \leftarrow \lambda$.          ▷ After the loop, $\lambda$ represents $\nabla J(x_N)^\top (\partial x_N/\partial x_0)$
11: **return** $S_\theta, S_{x_0}$.

---

**Proposition 1** (Discrete Adjoint Method). *Let the sequence $x_0, \ldots, x_N$ be generated by $x_{k+1} = x_k - h\mathcal{A}(\theta, x_k, t_k)$. The quantities $S_\theta$ and $S_{x_0}$ computed by Algorithm 1 are equal to $\nabla J(x_N)^\top (\partial x_N/\partial \theta)$ and $\nabla J(x_N)^\top (\partial x_N/\partial x_0)$, respectively.*

Once $S_\theta$ and $S_{x_0}$ are computed using Algorithm 1, they are plugged into expression (8). This approach computes the required numerators efficiently by only requiring storage for the forward trajectory $\{x_k\}$ and the current adjoint vector $\lambda$, making its memory footprint $O(Nd + d)$, which is typically much smaller than $O(N \times \text{memory for } \mathcal{A} \text{ graph})$ needed for naive unrolling. The computational cost is roughly proportional to $N$ times the cost of evaluating $\mathcal{A}$ and its relevant partial derivatives (or VJPs). The overall procedure to compute $\nabla_\theta N_J$ would first run the forward pass to find $N$ and store $\{x_k\}$, then call Algorithm 1 to get $S_\theta$, and finally assemble the components using (8).

## 3 Applications of Differentiable Stopping Time

In this section, we explore two applications of the differentiable discrete stopping time: L2O and online adaptation of learning rates (or other optimizer parameters). The ability to differentiate $N_J$ allows us to directly optimize for algorithmic efficiency towards target suboptimality.

### 3.1 L2O with Differentiable Stopping Time

In L2O, the objective is to learn an optimization algorithm, denoted by (2), parameterized by $\theta$, that performs efficiently across a distribution of optimization tasks. Traditional L2O approaches often aim to minimize a sum of objective function values over a predetermined number of steps. While this provides a dense reward signal, it may not directly optimize for the speed to reach a specific target precision $\varepsilon$. To overcome this limitation, the L2O training objective can be augmented with the stopping time

$$\min_\theta \quad \mathcal{L}(\theta) = \mathbb{E}_{f \sim \mathcal{D}_f, x_0 \sim \mathcal{D}_{x_0}} \left[ \sum_{k=0}^{K_{\max}} w_k f(x_k) + \lambda N_J(\theta, x_0, \varepsilon) \right], \tag{10}$$

where $\mathcal{D}_f, \mathcal{D}_{x_0}$ are distributions of $f$ and $x_0$, respectively, $K_{\max}$ is a maximum horizon for the sum, $J$ is a stopping criterion depending on $f$, $w_k$ are weights, $\lambda$ is a balancing hyperparameter, and $N_J(\theta, x_0, \varepsilon)$ is the discrete stopping time. The parameters $\theta$ are then updated using a stochastic

optimization method such as stochastic gradient descent or Adam. The update follows the rule

$$\theta_{\text{new}} = \theta_{\text{old}} - \eta_{\text{L2O}} \left( \nabla_\theta \left[ \sum_{k=0}^{K_{\max}} w_k f(x_k) \right] + \lambda \nabla_\theta N_J(\theta, x_0, \varepsilon) \right), \tag{11}$$

where $\eta_{\text{L2O}}$ is the meta-learning rate. Combining these two losses contributions provides a richer training signal that values both the quality of the optimization path and the overall convergence speed.

Suppose $f(x_k) > f(x_{k+1})$ holds for all $k$, another interesting result comes from the identity

$$\frac{\mathrm{d}}{\mathrm{d}\theta} \sum_{k=0}^{K_{\max}} f(x_k) = \sum_{k=0}^{K_{\max}} (f(x_k) - f(x_{k-1})) \frac{\nabla f(x_k) \partial x_k / \partial \theta}{f(x_k) - f(x_{k-1})}$$
$$= \frac{\partial}{\partial \theta} \sum_{k=0}^{K_{\max}} (f(x_{k-1}) - f(x_k)) N_f(\theta, x_{k-1}, f(x_k)). \tag{12}$$

We emphasize that the operator $\partial/\partial\theta$ directly applies to the variable $\theta$ while $\mathrm{d}/\mathrm{d}\theta$ will unroll the intermediate variable and apply chain rule. The identity (12) reveals that optimizing the weighted loss sum with $w_k \equiv 1$ equals to minimize the sum of stopping times greedily with stopping criterion $f$ and natural weights $f(x_{k-1}) - f(x_k)$.

## 3.2 Online Adaptation of Optimizer Parameters via Stopping Time

Online adaptation of optimizer hyperparameters $\theta_k$ (for $x_{k+1} = G(x_k, \theta_k)$) can be triggered by an adaptive criterion $\varphi(N, \varepsilon)$. This criterion, potentially adaptive itself, signals when to update $\theta_k$. $N$ is a stopping time from a reference $x_{\text{ref}}$ (last adaptation point or $x_0$) until $\varphi$ is met at $x_{\text{current}}$. Upon meeting $\varphi$ at $x_{k+1}(= x_{\text{current}})$, the sensitivity $\partial N / \partial \theta$ of the stopping time $N$ to hyperparameters $\theta$ active within $[x_{\text{ref}}, x_{k+1}]$ is key. Theoretically, $\partial N / \partial \theta$ is found by backpropagating through all steps from $x_{k+1}$ to $x_{\text{ref}}$, yielding a principled multi-step signal for adjusting $\theta$.

Calculating the full multi-step $\partial N / \partial \theta$ to $x_{\text{ref}}$ is often costly. Practical methods may truncate this dependency. The simplest truncation considers only the immediate impact of $\theta_k$ on $x_{k+1}$. For this single-step proxy $N_k$, its sensitivity, given $x_{k+1} = x_k - h_{\text{step}} \mathcal{A}(\theta_k, x_k, t_k)$ and decreasing $J(x)$, is

$$\frac{\partial N_k}{\partial \theta} = \frac{h_{\text{step}} \nabla J(x_{k+1})^\top (\partial \mathcal{A}(\theta_k, x_k, t_k) / \partial \theta)}{J(x_{k+1}) - J(x_k)}. \tag{13}$$

For Adam's learning rate $\alpha_k$ (where $x_{k+1} = x_k - \alpha_k d_k$, $\mathcal{A} = \alpha_k d_k$, $h_{\text{step}} = 1$, $\theta_k = \alpha_k$, and $\partial \mathcal{A} / \partial \alpha_k = d_k$), the one-step truncated sensitivity $S_k$ from (13) (with $J(x) = f(x)$) becomes

$$S_k = \frac{\nabla f(x_{k+1})^\top d_k}{f(x_{k+1}) - f(x_k)}. \tag{14}$$

$S_k$ is the practical signal for adjusting $\alpha_k$. If $f(x_{k+1}) < f(x_k)$ (negative denominator), $\alpha_k$ is updated by

$$\alpha_{k+1} = \alpha_k - \eta_{\text{adapt}} S_k, \tag{15}$$

with $\eta_{\text{adapt}}$ as the adaptation rate. Specifically, if $\nabla f(x_{k+1})^\top d_k > 0$, then $S_k < 0$, increasing $\alpha_k$; if $\nabla f(x_{k+1})^\top d_k < 0$, then $S_k > 0$, decreasing $\alpha_k$. The full procedure for Adam with Online LR Adaptation (Adam-OLA) is provided in Algorithm 2 in the Appendix D.

## 4 Experiments

**Validation of Theorems 2 and Proposition 1.** To validate the effectiveness and efficiency of our differentiable discrete stopping time approach, we conduct experiments on a high-dimensional quadratic optimization problem. We minimize $f(x) = x^\top Q x / 2$, $x \in \mathbb{R}^d$ with $d \in \{10^2, 10^3, 10^4\}$ and condition number 100. The optimization algorithm uses forward Euler discretization (2) of (1), where $\mathcal{A}$ incorporates a diagonal preconditioner (4) with $10d$ learnable parameters. The stopping criterion is $\|\nabla f(x)\|_2^2 \leq \varepsilon$ with $\varepsilon \in \{10^{-3}, 10^{-4}, 10^{-5}\}$. We compare the sensitivity of the discrete stopping time $\nabla_\theta N_J$, computed using Algorithm 1, against the gradient of the continuous stopping

time $\nabla_\theta T_J$ (ground truth), computed via `torchdiffeq` [27] through an adaptive ODE solver. We vary $d$, $\varepsilon$, and $h$.

We evaluate effectiveness and efficiency using two primary metrics, *Relative Error* quantifies the accuracy of $\nabla_\theta N_J$ as an approximation of $\nabla_\theta T_J$. A smaller error indicates better accuracy, with $\mathcal{O}(h)$ magnitude expected (Theorem 2). Results are shown in Figure 2a. *NFE Ratio* measures the computational cost efficiency, defined as the number of function evaluations (NFE) for Algorithm 1 to compute $\nabla_\theta N_J$ versus the adaptive ODE solver to compute $\nabla_\theta T_J$. A ratio $< 1$ indicates the discrete approach's forward simulation is cheaper. Results are shown in Figure 2b. The numbers of Euler NFE and ODE NFE are presented in Appendix D. The math formulae of these quantities are

$$\text{Relative Error} = \frac{\|\nabla_\theta N_J - \nabla_\theta T_J\|_2}{\|\nabla_\theta T_J\|_2 + \|\nabla_\theta N_J\|_2}, \qquad \text{NFE Ratio} = \frac{\text{Euler NFE}}{\text{ODE NFE}}.$$

By analyzing the relative error and NFE ratio across varying parameters, our experiments demonstrate that the discrete sensitivity provides an accurate approximation while requiring substantially fewer function evaluations for the forward pass, highlighting its efficiency and suitability for high-dimensional problems compared to methods relying on precise ODE solves for the stopping time gradient.

Notably, smaller stopping thresholds $\epsilon$ also lead to lower relative error. Intuitively, smaller values of $\epsilon$ lead to longer optimization trajectories that settle closer to the optimum, where the dynamics are smoother and the discrete approximation becomes more accurate. However, a deeper explanation is supported by the theoretical analysis in [25], which shows that $\|x_k - x(t_k)\|$ can decrease as $k$ increases, even under a fixed step size. This directly explains the trend: smaller $\epsilon$ leads to larger $k$, which in turn reduces the discrepancy between the discrete and continuous trajectories, and thus the relative error. This interpretation is also reinforced by Figure 1a. As the optimization progresses, the distance between the discrete iterates and the continuous path visibly decreases. In particular, the discrete and continuous trajectories gradually align as they approach the stopping region, further supporting the claim that gradient approximation becomes more accurate near convergence.

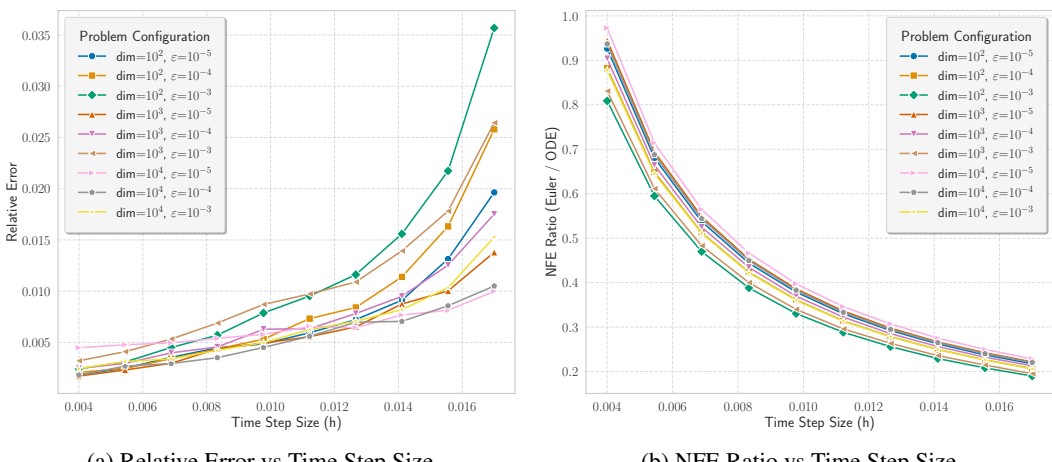

(a) Relative Error vs Time Step Size          (b) NFE Ratio vs Time Step Size

Figure 2: Experimental results comparing the discrete and continuous stopping time gradients across varying problem dimensions, stopping thresholds, and time step sizes. (a) shows the relative error of the discrete gradient approximation. (b) shows the computational cost ratio.

**Learning to Optimize.** We consider a logistic regression problem with synthetic data

$$\min_{x \in \mathbb{R}^d} \ f(x) \colon = \frac{1}{n} \sum_{i=1}^{n} \log\left(1 + \exp(-y_i w_i^\top x)\right),$$

where $w_i \in \mathbb{R}^d$ denotes the $i$-th data sample, and $y_i \in \{0, 1\}$ is the corresponding label. We consider two L2O optimizers: L2O-DM [28] and L2O-RNNprop [29]. Both employ a two-layer LSTM with a hidden state size of 30 to predict coordinate-wise updates. The data generation process and the architecture of the L2O optimizers are detailed in Appendix D. The training setup follows

that of [29]. Specifically, the feature dimension is set to $d = 512$, and the number of samples is $n = 256$. In each training step, we use a mini-batch consisting of 64 optimization problems. The total number of training steps is 500. In each of these steps, a batch of 64 optimization problems is sampled, and the learned optimizers are unrolled for a horizon of $K_{\max} = 100$ iterations to compute the training loss. We divide the sequence into 5 segments of 20 steps each and apply truncated backpropagation through time (BPTT) for training. The weights in (10) are set as $w_k \equiv 1/K_{\max}$. Two loss functions are considered. The first corresponds to setting $\lambda = 0$ in (10), resulting in an average loss across all iterations. To demonstrate the benefit of incorporating the stopping time penalty, we also set $\lambda = 1$ and use the stopping criterion $f(x_{k-1}) - f(x_k) \leq 10^{-5}$. This can be reformulated into the standard form by augmenting the state variable as $z_k = (x_k, x_{k-1})$ and defining $J(z) = f(z[d+1:2d]) - f(z[1:d])$.

The test results are summarized in Figure 3. `L2O-DM` refers to the L2O-DM optimizer. `L2O-RNNprop` and `L2O-RNNprop-Time` denote the L2O-RNNprop optimizer with and without the stopping time penalty, respectively. Since L2O-DM does not reach the stopping criterion within the maximum number of steps, we do not evaluate its performance under the stopping time penalty. For comparison with manually designed optimizers, `GD` represents gradient descent, `NAG` denotes Nesterov's accelerated gradient method, and `Adam` is a well-known adaptive optimizer. All classical methods use a fixed step size of $1/L$, where $L$ is the Lipschitz constant of $\nabla f(x)$ estimated at the initial point $x_0$. Our results show a clear acceleration toward reaching the target stopping criterion. In Figure 3a, we evaluate on a problem of the same size, $d = 512$, $n = 256$. In Figure 3b, we test on a fourfold larger instance with $d = 2048$, $n = 1024$. Both experiments indicate that the number of iterations required to meet the stopping criterion is reduced by hundreds of steps, and the learned optimizers generalize robustly to larger-scale problems.

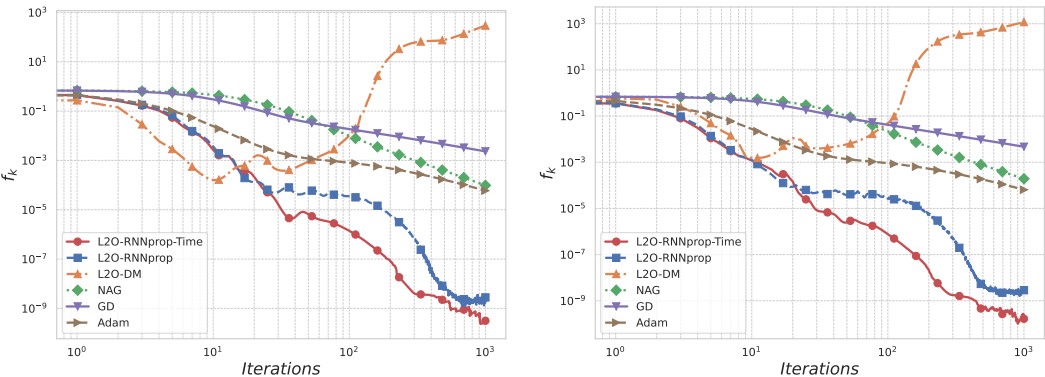

(a) Train and test on the same size problems.          (b) Test on 4x larger problems than training.

Figure 3: Test results of different optimizers on logistic regression: Function value versus iteration.

**Online Learning Rate Adaptation.** We tested Algorithm 2 on smooth support vector machine (SVM) problems [30], using datasets from LIBSVM [31]. `HB` denotes the heavy-ball method, and `NAG-SC` refers to the Nesterov accelerated gradient method tailored for strongly convex objectives. `Adagrad` is an adaptive gradient algorithm that scales the learning rate per coordinate based on historical gradient information. `Adam-HD` is an influential extension of Adam [32] in the context of online learning rate adaptation; it updates the base learning rate of `Adam` at each iteration using a hyper-gradient technique. The remaining abbreviations retain their previously defined meanings. The results presented in Figure 4 demonstrate that Algorithm 2 consistently outperforms the baseline methods, particularly in the later stages of convergence. Further comparisons across multiple datasets, as well as detailed descriptions of hyperparameter settings for the baselines, are provided in Appendix D.

## 5 Conclusion

In this work, we introduced the concept of a differentiable discrete stopping time for iterative algorithms, establishing a link between continuous time dynamics and their discrete approximations. We proposed an efficient method using the discrete adjoint principle to compute the sensitivity of the discrete stopping time. Our experiments demonstrate that this approach provides an accurate

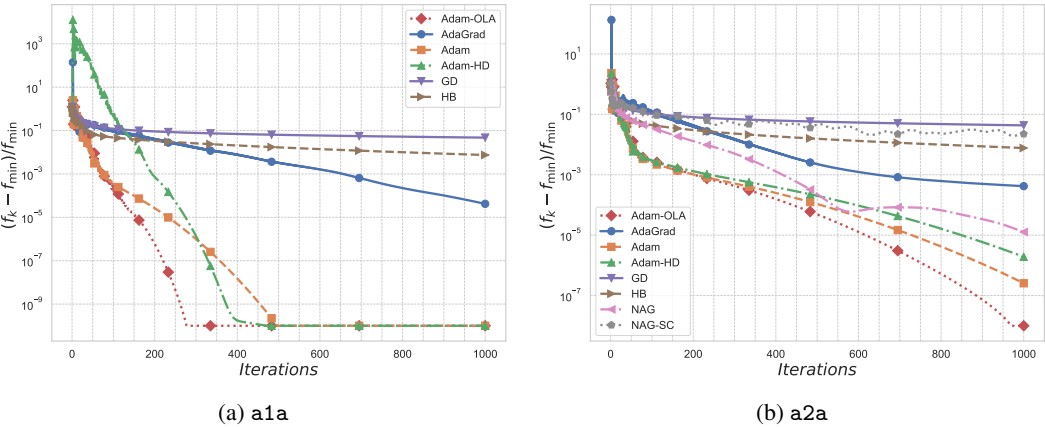

Figure 4: Comparison of different optimizers on smooth SVM: Function value versus iteration. Here, $f_{\min}$ denotes the minimum function value achieved across all iterations for each optimizer.

gradient approximation while requiring substantially fewer function evaluations for the forward pass compared to methods relying on continuous ODE solves, proving efficient and scalable for high-dimensional problems. This allows direct optimization of algorithms for convergence speed, with potential applications in L2O and online adaptation.

However, we note that employing a forward Euler discretization with a fixed time step may be too coarse for the algorithmic design. This limitation is also reflected in the error bound estimated in Theorem 2. In future work, we plan to explore more tailored algorithmic designs for $\mathcal{A}$ alongside more sophisticated discretization schemes—such as symplectic integrators or methods that incorporate higher-order information. Such approaches may enable more accurate control of the global error and allow for a wider range of stable time steps during discretization.

## Acknowledgement

This research was supported in part by the National Natural Science Foundation of China under the grant numbers 12331010 and 12288101, National Key Research and Development Program of China under the grant number 2024YFA1012902, and the Natural Science Foundation of Beijing, China under the grant number Z230002.

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

# A    Proof of Theorem 1

*Proof.* Consider a function $G(\theta, x_0, t) = J(x(t; \theta, x_0)) - \epsilon$. By the definition of $T = T_J(\theta, x_0, \epsilon)$, we have

$$G(\theta, x_0, T) = G(\theta, x_0, T_J(\theta, x_0, \epsilon)) = 0.$$

Computing the partial derivatives yields

$$\frac{\partial G}{\partial \theta} = \nabla J(x)^\top \frac{\partial x}{\partial \theta},$$

and

$$\frac{\partial G}{\partial t} = \nabla J(x)^\top \dot{x}(t).$$

Using the implicit function theorem, we conclude that $T$ can be expressed locally as a continuously differentiable function of $\theta$ or $x_0$. We now differentiate $G$ with respect to $\theta$ and $x_0$, which yields

$$0 = \frac{\mathrm{d}}{\mathrm{d}\theta}\left(G(\theta, x_0, T)\right) = \frac{\mathrm{d}}{\mathrm{d}\theta} J\left(x(T(\theta, x_0, \epsilon); \theta, x_0)\right) = \nabla J(x)^\top \left(\frac{\partial x}{\partial \theta} + \dot{x}(T)\frac{\partial T}{\partial \theta}\right),$$

and

$$0 = \frac{\mathrm{d}}{\mathrm{d}x_0}\left(G(\theta, x_0, T)\right) = \frac{\mathrm{d}}{\mathrm{d}x_0} J\left(x(T(\theta, x_0, \epsilon); \theta, x_0)\right) = \nabla J(x)^\top \left(\frac{\partial x}{\partial x_0} + \dot{x}(T)\frac{\partial T}{\partial x_0}\right).$$

Rearranging these equations leads to

$$\nabla J(x)^\top \dot{x}(T)\frac{\partial T}{\partial \theta} = -\nabla J(x)^\top \frac{\partial x}{\partial \theta}, \quad \text{and hence} \quad \frac{\partial T}{\partial \theta} = \left(\nabla J(x)^\top \dot{x}(T)\right)^{-1} \nabla J(x)^\top \frac{\partial x}{\partial \theta},$$

as well as

$$\nabla J(x)^\top \dot{x}(T)\frac{\partial T}{\partial x_0} = -\nabla J(x)^\top \frac{\partial x}{\partial x_0}, \quad \text{and hence} \quad \frac{\partial T}{\partial x_0} = \left(\nabla J(x)^\top \dot{x}(T)\right)^{-1} \nabla J(x)^\top \frac{\partial x}{\partial x_0}.$$

The above equations complete the proof. $\qquad\square$

# B    Proof of Theorem 2

We first present a basic analysis in numerical ODEs.

**Proposition 2** (Error analysis of the forward Euler method). *Let $f : \mathbb{R}^n \times \mathbb{R} \to \mathbb{R}$ be a function defined by $(x, t) \mapsto f(x, t)$. Suppose the following assumptions hold*

1. *There exists a constant $L_x > 0$ such that $\|f(x_1, t) - f(x_2, t)\| \leq L_x\|x_1 - x_2\|$ for all $x_1, x_2$, and $t$.*

2. *There exists a constant $L_t > 0$ such that $\|f(x, t_1) - f(x, t_2)\| \leq L_t|t_1 - t_2|$ for all $x, t_1$, and $t_2$.*

3. *There exists a constant $M > 0$ such that $\|f(x, t)\| < M$ for all $x$ and $t$.*

*Given an initial condition $x(t_0) = x_0$ and a fixed stepsize $h$, we consider the sequence generated by the forward Euler method as*

$$x_{k+1} = x_k + hf(x_k, t_k), \quad t_k = t_0 + kh.$$

*Then, for any positive integer $k$, the error $e_k = x_k - x(t_k)$ satisfies*

$$\|e_k\| \leq \frac{h}{2}\left(M + \frac{L_t}{L_x}\right)\left(e^{L_x hk} - 1\right).$$

*Proof.* We begin by expressing the error at step $k + 1$ as

$$e_{k+1} = x_{k+1} - x(t_{k+1}) = e_k + h\left(f(x_k, t_k) - f(x(t_k), t_k)\right) + x(t_k) + hf(x(t_k), t_k) - x(t_{k+1}).$$

Applying Lipschitz continuity, we obtain the inequality

$$\|e_{k+1}\| \le (1 + L_x h)\|e_k\| + \|x(t_k) + h f(x(t_k), t_k) - x(t_{k+1})\|.$$

The second term on the right-hand side can be expressed in integral form as

$$\|x(t_k) + h f(x(t_k), t_k) - x(t_{k+1})\| = \left\| \int_{t_k}^{t_{k+1}} [f(x(t), t) - f(x(t_k), t_k)]\, dt \right\|,$$

which is bounded above by

$$\left\| \int_{t_k}^{t_{k+1}} [f(x(t), t_k) - f(x(t_k), t_k)]\, dt \right\| + \left\| \int_{t_k}^{t_{k+1}} [f(x(t), t) - f(x(t), t_k)]\, dt \right\|.$$

Substituting the assumptions, we estimate the first integral as

$$\left\| \int_{t_k}^{t_{k+1}} [f(x(t), t_k) - f(x(t_k), t_k)]\, dt \right\| \le L_x \int_{t_k}^{t_{k+1}} \|x(t) - x(t_k)\| dt \le \frac{1}{2} M L_x h^2,$$

where the last inequality follows from the Lagrange mean value theorem, which implies that

$$\int_{t_k}^{t_{k+1}} \|x(t) - x(t_k)\| dt = \int_{t_k}^{t_{k+1}} \|\dot{x}(\xi)\| |t - t_k| dt$$

$$= \int_{t_k}^{t_{k+1}} \|f(x(\xi), \xi)\| |t - t_k| dt$$

$$\le M \int_{t_k}^{t_{k+1}} |t - t_k| dt = \frac{1}{2} M h^2.$$

Similarly, for the second integral, we derive the bound as

$$\left\| \int_{t_k}^{t_{k+1}} [f(x(t), t) - f(x(t), t_k)]\, dt \right\| \le L_t \int_{t_k}^{t_{k+1}} |t - t_k| dt = \frac{1}{2} L_t h^2.$$

Combining these inequalities, we obtain

$$\|e_{k+1}\| \le (1 + L_x h)\|e_k\| + \frac{h^2}{2}(L_t + M L_x).$$

Finally, using the initial error $e_0 = 0$, we conclude that the global error satisfies

$$\|e_k\| \le \frac{h}{2}\left(M + \frac{L_t}{L_x}\right)\left(e^{L_x h k} - 1\right).$$

This completes the proof. □

*Proof of the Theorem.* The following conditions are assumed throughout our analysis. First, the function $\mathcal{A}$ is twice continuously differentiable, i.e., $\mathcal{A} \in C^2$. Second, $\mathcal{A}$ itself, together with all partial derivatives of $\mathcal{A}$ (such as $\frac{\partial}{\partial x}\mathcal{A}$, $\frac{\partial^2 \mathcal{A}}{\partial \theta \partial t}$) and the gradient and Hessian of $J$ (i.e., $\nabla J$ and $\nabla^2 J$), are uniformly bounded by constants $A, A_x, A_\theta, A_t, A_{\theta,x}, A_{x,x}, A_{x,t}, A_{\theta,t}, J_1$, and $J_2$, respectively. Third, we assume the boundary condition $|\nabla J(x(T))^\top \dot{x}(T)| = \delta$.

For clarity and brevity, our main theorem states the regularity and boundedness assumptions using Sobolev norms; specifically, we require that $\mathcal{A}(\theta, x(t), t)$, regarded as a function of $(\theta, t)$, has uniformly bounded $W^{2,\infty}$ norms with respect to $(\theta, t)$ in a neighborhood of $\theta \times [t_0, \ t_0 + N_J h]$, and that $J$, regarded as a function of $x$, has a uniformly bounded $W^{2,\infty}$ norm in a neighborhood of $x(T_J)$. In this proof, we equivalently expand these assumptions by explicitly introducing uniform bounds for $\mathcal{A}$, its partial derivatives (with respect to $x$, $\theta$, $t$, etc.), and for the gradient $\nabla J$ and Hessian $\nabla^2 J$, denoted by $A, A_x, A_\theta, A_t, A_{\theta,x}, A_{x,x}, A_{x,t}, A_{\theta,t}, J_1$, and $J_2$, respectively. This explicit formulation is purely for notational convenience in the analysis, as it allows us to refer directly to these quantities

in the derivations, especially during Taylor expansions and error estimates. We emphasize that these detailed bounds can be derived from the $W^{2,\infty}$ norm boundedness assumed in the theorem statement.

Without loss of generality, we only prove the case for the $L^2$ norm. We first recall the form and the definition of the derivative. They are given by

$$\nabla_\theta T = \nabla_\theta T_J(\theta, x_0, \epsilon) = -\frac{\nabla J(x(T))^\top \frac{\partial x(T)}{\partial \theta}}{\nabla J(x(T))^\top \dot{x}(T)},$$

$$\nabla_\theta N = \nabla_\theta N_J(\theta, x_0, \epsilon) = -\frac{h \nabla J(x_N)^\top \frac{\partial x_N}{\partial \theta}}{J(x_N) - J(x_{N-1})}.$$

We consider the iteration

$$x_{k+1} = x_k - h\mathcal{A}(\theta, x_k, t_k).$$

Differentiating with respect to $\theta$, we obtain

$$\frac{\partial x_{k+1}}{\partial \theta} = \left( I - h\frac{\partial}{\partial x}\mathcal{A}(\theta, x_k, t_k) \right) \frac{\partial x_k}{\partial \theta} - h\frac{\partial}{\partial \theta}\mathcal{A}(\theta, x_k, t_k),$$

where the initial condition $\frac{\partial x_0}{\partial \theta}$ holds.

Also, we consider the flow

$$\dot{x}(t) = -\mathcal{A}(\theta, x(t), t).$$

Differentiating with respect to $\theta$, we obtain

$$\frac{\mathrm{d}}{\mathrm{d}t}\frac{\partial x(t)}{\partial \theta} = -\frac{\partial}{\partial \theta}\mathcal{A}(\theta, x(t), t) - \frac{\partial}{\partial x}\mathcal{A}(\theta, x(t), t)\frac{\partial x(t)}{\partial \theta}, \tag{16}$$

where the initial condition $\frac{\partial x(t_0)}{\partial \theta} = 0$ holds. Let $u(t) = \frac{\partial x(t)}{\partial \theta}$. It is easy to observe that the iteration above corresponds to the forward Euler method for solving the ODE

$$\frac{\mathrm{d}}{\mathrm{d}t}u(t) = -\frac{\partial}{\partial \theta}\mathcal{A}(\theta, x(t), t) - \frac{\partial}{\partial x}\mathcal{A}(\theta, x(t), t)u(t).$$

We now proceed to show that $u(t)$, for $t \in [t_0, T]$, is bounded by some constant $M > 0$. Let $v = u^\top u$, $B(t) = -\frac{\partial}{\partial \theta}\mathcal{A}(\theta, x(t), t)$, and $C(t) = -\frac{\partial}{\partial x}\mathcal{A}(\theta, x(t), t)$. Then we can derive that

$$\frac{\mathrm{d}}{\mathrm{d}t}v = 2u^\top \frac{\mathrm{d}}{\mathrm{d}t}u = 2u^\top B + 2u^\top C u.$$

Therefore, we have the bound

$$\left| \frac{\mathrm{d}}{\mathrm{d}t}v \right| \le 2\|B\|\sqrt{v} + 2\|C\|v \le \|B\| + (\|B\| + 2\|C\|)v \le A_\theta + (A_\theta + 2A_x)v.$$

Applying the Gronwall inequality, for every $t \in [t_0, T]$, we obtain the following estimate

$$\|u(t)\| = \sqrt{v(t)} \le \sqrt{\frac{A_\theta}{A_\theta + 2A_x}\left(e^{(A_\theta + 2A_x)(T-t_0)} - 1\right)} \triangleq M. \tag{17}$$

Employing Proposition 2, we obtain that $\left\| \frac{\partial x_N}{\partial \theta} - \frac{\partial x(Nh)}{\partial \theta} \right\|$ is bounded by

$$\frac{h}{2}\left( MA_x + A_\theta + \frac{M(A_{x,t} + AA_{x,x}) + A_{\theta,t} + AA_{\theta,x}}{A_x} \right)\left( e^{A_x(T+1-t_0)} - 1 \right).$$

Let

$$c_1 \triangleq \frac{1}{2}\left( MA_x + A_\theta + \frac{M(A_{x,t} + AA_{x,x}) + A_{\theta,t} + AA_{\theta,x}}{A_x} \right)\left( e^{A_x(T+1-t_0)} - 1 \right). \tag{18}$$

Noticing that $\frac{\mathrm{d}}{\mathrm{d}t}\frac{\partial x(t)}{\partial \theta}$ is bounded by $A_\theta + A_x M$ according to (16), and that $|T - Nh| \le h$, we deduce that

$$\left\| \frac{\partial x(Nh)}{\partial \theta} - \frac{\partial x(T)}{\partial \theta} \right\| \le (A_\theta + A_x M)h.$$

Let $e_1 = \frac{\partial x_N}{\partial \theta} - \frac{\partial x(T)}{\partial \theta}$. Then we obtain the estimate

$$\|e_1\| \leq (A_\theta + A_x M + c_1)h. \tag{19}$$

Similarly, by Proposition 2, we know that

$$\|x_N - x(Nh)\| \leq \frac{h}{2}\left(A + \frac{A_t}{A_x}\right)\left(e^{A_x(T+1-t_0)} - 1\right).$$

Let

$$c_2 \triangleq \frac{1}{2}\left(A + \frac{A_t}{A_x}\right)\left(e^{A_x(T+1-t_0)} - 1\right). \tag{20}$$

Since $\frac{\mathrm{d}}{\mathrm{d}t}x(t) = -\mathcal{A}(\theta, x(t), t)$ is bounded by $A$ and $|T - Nh| \leq h$, it follows that

$$\|x_N - x(T)\| \leq (A + c_2)h.$$

Let $e_2 = \nabla J(x_N) - \nabla J(x(T))$. Since $\|\nabla^2 J\|$ is bounded by $J_2$, we obtain the estimate

$$|e_2| \leq J_2(A + c_2)h. \tag{21}$$

The Taylor expansion yields

$$J(x_N) = J(x_{N-1}) - \nabla J(x_{N-1})^\top(x_N - x_{N-1}) + \frac{1}{2}(x_N - x_{N-1})^\top \nabla^2 J(\xi)(x_N - x_{N-1}).$$

Combining this with the fact that $x_N - x_{N-1} = -h\mathcal{A}(\theta, x_{N-1}, t_{N-1})$ and that $\|\nabla^2 J\|$ is bounded by $J_2$, we obtain

$$\left|\frac{J(x_N) - J(x_{N-1})}{h} + \nabla J(x_{N-1})^\top \mathcal{A}(\theta, x_{N-1}, t_{N-1})\right| \leq \frac{1}{2}hA^2 J_2.$$

Let

$$e_5 = \frac{J(x_N) - J(x_{N-1})}{h} + \nabla J(x_{N-1})^\top \mathcal{A}(\theta, x_{N-1}, t_{N-1})$$

and $e_3 = \nabla J(x_{N-1}) - \nabla J(x(T))$, $e_4 = \mathcal{A}(\theta, x_{N-1}, t_{N-1}) + \dot{x}(T)$. We have just derived

$$|e_5| \leq \frac{1}{2}hA^2 J_2 \tag{22}$$

As in the previous estimate for $e_2$, we obtain

$$\|e_3\| \leq J_2(A + c_2)h. \tag{23}$$

Furthermore, we have

$$\|e_4\| \leq \|\mathcal{A}(\theta, x_{N-1}, t_{N-1}) - \mathcal{A}(\theta, x(T), t_{N-1})\| + \|\mathcal{A}(\theta, x(T), t_{N-1}) - \mathcal{A}(\theta, x(T), T)\|$$
$$\leq A_x(A + c_2)h + A_t h. \tag{24}$$

Substituting the definitions of these error terms into the expression for $\nabla_\theta N$, we obtain

$$\nabla_\theta N = -\frac{(\nabla J(x(T)) + e_2)^\top \left(\frac{\partial x(T)}{\partial \theta} + e_1\right)}{(\nabla J(x(T)) + e_3)^\top (\dot{x}(T) - e_4) + e_5}.$$

Recall that

$$\nabla_\theta T = -\frac{\nabla J(x(T))^\top \frac{\partial x(T)}{\partial \theta}}{\nabla J(x(T))^\top \dot{x}(T)}.$$

Comparing these two expressions and combining the estimates from (19), (21), (23), and (24), together with the assumptions, we arrive at the final estimate that

$$\|\nabla_\theta T - \nabla_\theta N\| \leq Rh + \mathcal{O}(h^2),$$

where

$$R = \frac{J_1 M}{\delta^2}\left(J_1(A_t + A_x(A + c_2)) + \frac{3}{2}A^2 J_2 + AJ_2 c_2\right)$$

$$+ \frac{1}{\delta}\left(J_1(A_0 + A_x M + c_1) + MJ_2(A + c_2)\right).$$

Here, the constants refer to those defined in (17), (18), (20), and the assumptions stated earlier. This completes the proof. $\qquad\square$

## C   Proof of Proposition 1

*Proof.* We aim to compute $S_{\theta_j} = \nabla J(x_N)^\top \frac{\partial x_N}{\partial \theta_j}$ for each component $\theta_j$ of $\theta$, and $S_{x_0} = \nabla J(x_N)^\top \frac{\partial x_N}{\partial x_0}$. Let $L(\theta, x_0) = J(x_N(\theta, x_0))$. We are interested in $\nabla_\theta L$ and $\nabla_{x_0} L$. Define the adjoint (co-state) vectors $\lambda_k \in \mathbb{R}^d$ for $k = 0, \dots, N$ such that $\lambda_k^\top = \frac{\partial J(x_N)}{\partial x_k} = \nabla J(x_N)^\top \frac{\partial x_N}{\partial x_k}$. The base case is at $k = N$,

$$\lambda_N = \frac{\partial J(x_N)}{\partial x_N} = \nabla J(x_N). \tag{25}$$

For $k < N$, $x_N$ depends on $x_k$ through $x_{k+1}$. Using the chain rule

$$\frac{\partial J(x_N)}{\partial x_k} = \frac{\partial J(x_N)}{\partial x_{k+1}} \frac{\partial x_{k+1}}{\partial x_k}.$$

In terms of our adjoints, we have

$$\lambda_k^\top = \lambda_{k+1}^\top \frac{\partial x_{k+1}}{\partial x_k}.$$

Given $x_{k+1} = x_k - h\mathcal{A}(\theta, x_k, t_k)$, the Jacobian is $\frac{\partial x_{k+1}}{\partial x_k} = I - h\frac{\partial \mathcal{A}(\theta, x_k, t_k)}{\partial x_k}$. Thus, the backward recursion for the adjoints is

$$\lambda_k^\top = \lambda_{k+1}^\top \left( I - h\frac{\partial \mathcal{A}(\theta, x_k, t_k)}{\partial x_k} \right), \tag{26}$$

or $\lambda_k = \left( I - h\frac{\partial \mathcal{A}(\theta, x_k, t_k)}{\partial x_k} \right)^\top \lambda_{k+1}$. The loop in Algorithm 1 implements this recursion. At the beginning of iteration $k$ (loop index in algorithm, representing the step from $x_k$ to $x_{k+1}$), the variable $\lambda$ in the algorithm holds $\lambda_{k+1}$ from our derivation.

Now consider the derivative with respect to a parameter $\theta_j$. $J(x_N)$ depends on $\theta_j$ through all $x_m$ for $m \leq N$ where $x_m$ is influenced by $\theta_j$. Hence,

$$\frac{\partial J(x_N)}{\partial \theta_j} = \sum_{m=0}^{N-1} \frac{\partial J(x_N)}{\partial x_{m+1}} \left( \frac{\partial x_{m+1}}{\partial \theta_j} \right)_{\text{explicit}},$$

where $(\partial x_{m+1}/\partial \theta_j)_{\text{explicit}}$ means differentiating $x_{m+1} = x_m - h\mathcal{A}(\theta, x_m, t_m)$ with respect to $\theta_j$ while holding $x_m$ fixed

$$\left( \frac{\partial x_{m+1}}{\partial \theta_j} \right)_{\text{explicit}} = -h\frac{\partial \mathcal{A}(\theta, x_m, t_m)}{\partial \theta_j}.$$

Thus, it holds

$$\frac{\partial J(x_N)}{\partial \theta_j} = \sum_{m=0}^{N-1} \lambda_{m+1}^\top \left( -h\frac{\partial \mathcal{A}(\theta, x_m, t_m)}{\partial \theta_j} \right). \tag{27}$$

The loop runs from $k = N - 1$ down to 0. For each $k$ in the loop, the term added is $-h(\frac{\partial \mathcal{A}(\theta, x_k, t_k)}{\partial \theta})^\top \lambda_{k+1}$. Summing these terms gives $\left( \nabla J(x_N)^\top \frac{\partial x_N}{\partial \theta} \right)_j$.

Finally, for the sensitivity with respect to $x_0$,

$$\frac{\partial J(x_N)}{\partial x_0} = \lambda_0^\top.$$

After the loop in Algorithm 1 finishes (i.e., after the iteration for $k = 0$), the variable $\lambda$ will have been updated using $\lambda_1$ and $\frac{\partial \mathcal{A}(\theta, x_0, t_0)}{\partial x_0}$, thus holding $\lambda_0$. $\square$

# D   Details of Experiments

---

**Algorithm 2** Adam-OLA

---

1: **Input:** $x_0, \alpha_0, f, \nabla f$.
2: **Params:** $\beta_1, \beta_2, \varepsilon_{\text{stab}}, \eta_{\text{adapt}}, \epsilon_{\text{desc}}$.
3: $m_0, v_0 \leftarrow 0, 0; k \leftarrow 0; \alpha_{\text{curr}} \leftarrow \alpha_0$.
4: $x_{\text{ref}} \leftarrow x_0; f_{\text{ref}} \leftarrow f(x_0); N_{\text{updates}} \leftarrow 0$.
5: **for** $k = 0, 1, \ldots$ until convergence **do**
6:     $g_k \leftarrow \nabla f(x_k)$
7:     $m_{k+1} \leftarrow \beta_1 m_k + (1 - \beta_1) g_k$
8:     $v_{k+1} \leftarrow \beta_2 v_k + (1 - \beta_2) g_k^2$
9:     $\hat{m}_{k+1} \leftarrow m_{k+1}/(1 - \beta_1^{k+1})$
10:     $\hat{v}_{k+1} \leftarrow v_{k+1}/(1 - \beta_2^{k+1})$
11:     $d_k \leftarrow \hat{m}_{k+1}/(\sqrt{\hat{v}_{k+1}} + \varepsilon_{\text{stab}})$
12:     $f_k^{\text{prev}} \leftarrow f(x_k)$
13:     $x_{k+1} \leftarrow x_k - \alpha_{\text{curr}} d_k$
14:     $f_{k+1} \leftarrow f(x_{k+1})$
15:     **if** $f_{\text{ref}} - f_{k+1} > \epsilon_{\text{desc}} \cdot N_{\text{updates}}$ **then**
16:         $g_{k+1}^{\text{new}} \leftarrow \nabla f(x_{k+1})$
17:         $\Delta f_{\text{step}} \leftarrow f_{k+1} - f_k^{\text{prev}}$
18:         **if** $\Delta f_{\text{step}} \neq 0$ **then**
19:             $S_k \leftarrow (g_{k+1}^{\text{new}} \cdot d_k)/\Delta f_{\text{step}}$
20:             $\alpha_{\text{curr}} \leftarrow \alpha_{\text{curr}} - \eta_{\text{adapt}} S_k$
21:         **end if**
22:         $x_{\text{ref}} \leftarrow x_{k+1}; f_{\text{ref}} \leftarrow f_{k+1}$
23:         $N_{\text{updates}} \leftarrow N_{\text{updates}} + 1$
24:     **end if**
25: **end for**
26: **Return** $x_k$

---

**Implementation Details.** We adopt the official implementation of [33][1] for the online learning rate adaptation experiments, and the codebase from [34][2] for L2O experiments. They all follow the MIT License as specified in their respective GitHub repositories. All experiments are conducted on a workstation running Ubuntu with a 12-core Intel Xeon Platinum 8458P CPU (2.7GHz, 44 threads), one NVIDIA RTX 4090 GPU with 24GB memory, and 60GB of RAM. We note that, for both experimental setups, we have made moderate modifications to the original implementations to better align with the goals of our study. However, as the focus of this work is to explore the potential applications of stopping time in optimization rather than to achieve state-of-the-art performance across all settings, we did not perform extensive hyperparameter tuning for the stopping time–based algorithms under different configurations. This choice may explain why our method does not reach SOTA performance in some scenarios.

**NFEs of different solvers.** Figure 5 shows that the NFE for an adaptive solver is mainly influenced by the stopping criterion. Since it does not accept a prespecified time step size, all of the statistics remain the same for different $h$.

**Hyperparameters of Baselines.** `Adagrad` is an adaptive gradient algorithm that adjusts learning rates per coordinate based on historical gradient information. The learning rate is set $\beta \in \{10^{-3}, 10^{-2}, 10^{-1}, 1.0, 10.0, 1/L\}$ with $\epsilon = 10^{-8}$. For Heavy-Ball method (HB), the momentum parameter is selected from the set $\{0.1, 0.5, 0.9, 1.0\}$. `Adam-HD` is a notable variant of Adam [32], which employs a hypergradient-based scheme to adaptively update the base learning rate at each iteration in an online fashion. For `Adam-HD`, the hyperparameter $\beta$ used to update the learning rate is chosen from the set $\{10^{-3}, 10^{-4}, 10^{-5}, 10^{-6}\}$. All other abbreviations follow their previously defined roles within the L2O framework. `Adam-OLA` and `Adam-HD` are all based on the

---

[1] https://github.com/udellgroup/hypergrad
[2] https://github.com/xhchrn/MS4LO

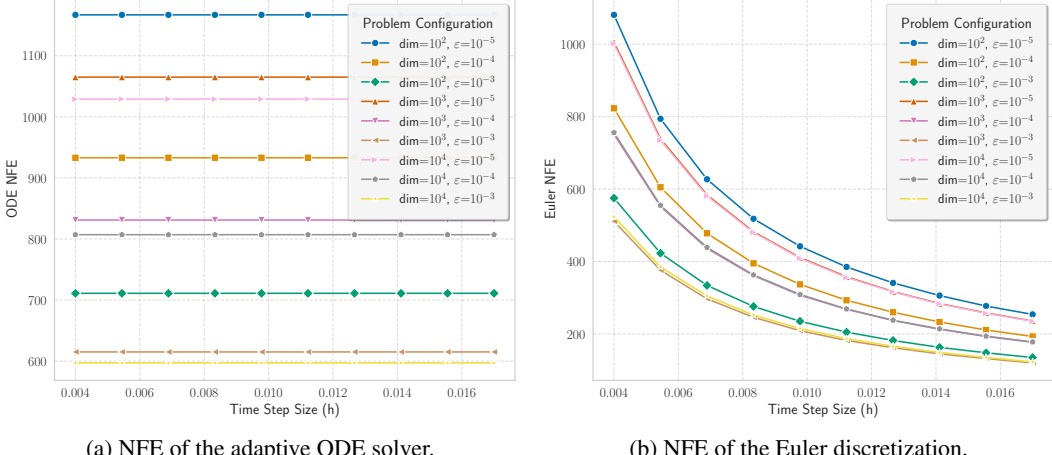

(a) NFE of the adaptive ODE solver.  (b) NFE of the Euler discretization.

Figure 5: NFEs of different solvers.

classical `Adam`, where $(\beta_1, \beta_2) = (0.9, 0.999)$ and $\epsilon = 10^{-8}$. The initial learning rate for Adam is selected from the set $\alpha \in \{10^{-3}, 10^{-2}, 10^{-1}, 1.0, 10.0, 1/L\}$. $L$ is the Lipschitz constant of $\nabla f(x)$, estimated at the initial point $x_0$. The maximum number of iterations is set to 1000, with a stopping criterion tolerance of $10^{-4}$.

Table 1: Hyperparameter settings for `Adam-OLA` on different datasets. The parameter $\beta$ controls the learning rate adaptation magnitude, and $\epsilon$ specifies the sufficient decrease threshold for triggering a learning rate update.

| Dataset (Experiment) | $\beta$ (Learning Rate Update) | $\epsilon$ (Descent Threshold) |
|---|---|---|
| a1a (*exp_svm*) | $1 \times 10^{-2}$ | $1 \times 10^{-5}$ |
| a2a (*exp_svm*) | $1 \times 10^{-3}$ | $1 \times 10^{-3}$ |
| a3a (*exp_svm*) | $5 \times 10^{-5}$ | $5 \times 10^{-4}$ |
| w3a (*exp_svm*) | $0.005$ | $5 \times 10^{-9}$ |

**Formulation of the Smooth SVM.** In this work, we consider the problem of binary classification using a smooth variant of the SVM, where the non-smooth hinge loss is replaced by its squared counterpart to enable efficient gradient-based optimization. Given a dataset $\{(x_i, y_i)\}_{i=1}^n$ with feature vectors $x_i \in \mathbb{R}^d$ and binary labels $y_i \in \{-1, +1\}$, the objective function takes the form

$$f(w) = \frac{1}{2} \sum_{i=1}^n \left[\max(0, 1 - y_i w^\top x_i)\right]^2 + \frac{\lambda}{2} \|w\|^2,$$

where $\lambda > 0$ is a regularization parameter. This formulation preserves the margin-maximizing behavior of the original SVM while allowing for stable and differentiable optimization. We further incorporate an intercept term into the model by appending a constant feature to each input vector. The resulting problem is solved using first-order methods with step size determined via an estimate of the gradient's Lipschitz constant.

**More Examples of Online Learning Rate Adaptation.** We report the performance of Algorithm 2 and other baseline methods. Our method shows consistent improvement in the later stage of the convergence.

**Data Synthetic Setting for L2O.** The data is synthetically generated. We first sample a sparse ground truth vector $x^\star \in \mathbb{R}^d$ with a prescribed sparsity level $s$, and then sample $W \in \mathbb{R}^{n \times d}$ with standard normal entries. The binary labels are generated via

$$y_i = \mathbf{1}_{\{w_i^\top x^\star \geq 0\}}, \quad i = 1, \ldots, n.$$

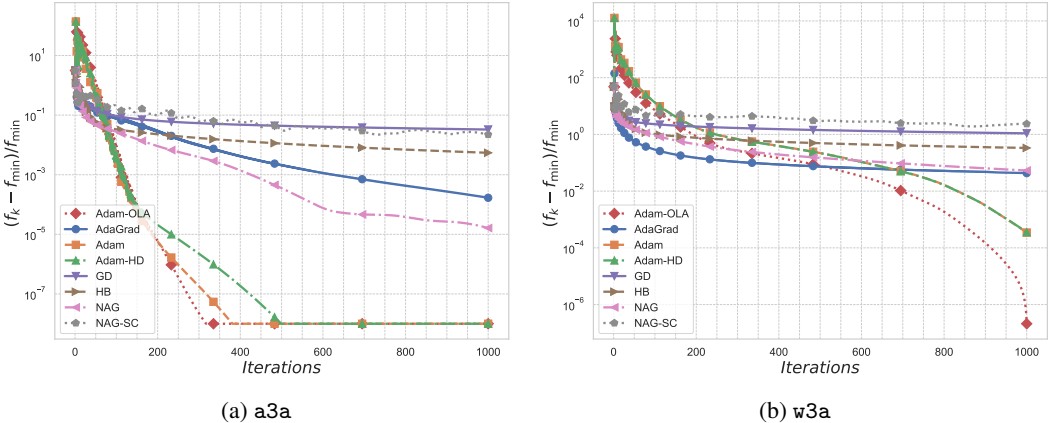

(a) `a3a`  (b) `w3a`

Figure 6: Comparison of different optimizers on smooth SVM: Function value versus iteration. Here, $f_{\min}$ denotes the minimum function value achieved across all iterations for each optimizer.

A small proportion of labels are flipped to simulate noise.

**Architectures of L2O Optimizers.** We now provide two examples of learned optimizers formulated within this framework, drawing from seminal works in the field. These learned optimizers typically output a direct parameter update $U_k$ such that $x_{k+1} = x_k + U_k$. To fit the continuous-time dynamical system framework where $x_{k+1} = x_k - h\mathcal{A}(\mathbf{w}_{opt}, x_k, t_k)$, we define $\mathcal{A}(\mathbf{w}_{opt}, x_k, t_k) = -U_k/h$. Here, $\mathbf{w}_{opt}$ denotes the parameters of the learned optimizer itself, $x_k$ are the parameters being optimized, and $h$ is the discretization step size from the underlying ODE.

**Detailed L2O Training Procedure.** The training of the L2O optimizers follows the paradigm described in [29]. The goal is to learn the parameters $\theta$ of the optimizer by minimizing the expected loss defined in Equation (10). The training process consists of 500 steps. In each step, we sample a mini-batch of 64 distinct logistic regression problems. For each problem in the batch, we unroll the learned optimizer for $K_{\max} = 100$ iterations, starting from a random initialization $x_0$. The loss for that problem is computed based on the trajectory $\{x_k\}_{k=0}^{100}$ according to Equation (10). To manage memory and computational cost, we use Truncated Backpropagation Through Time (BPTT), dividing the 100-step trajectory into 5 segments of 20 steps each. Gradients are computed for each segment and then averaged. The final gradient for the parameters $\theta$ is the average of the gradients computed across all 64 problems in the mini-batch. This gradient is then used to update $\theta$ with the Adam optimizer.

**LSTM-based Optimizer.** The influential work by Andrychowicz et al. [28] introduced an optimizer based on a Long Short-Term Memory (LSTM) network, which we denote as $m_{\mathbf{w}_{opt}}$. This optimizer operates coordinate-wise, meaning a small, shared-weight LSTM is applied to each parameter (coordinate) of the function $f(x)$ being optimized. For each coordinate, the LSTM takes the corresponding component of the gradient $\nabla f(x(t))$ and its own previous state, state$(t)$, as input to compute the parameter update component $U(t) = m_{\mathbf{w}_{opt}}(\nabla f(x(t)), \text{state}(t))$. The term state$(t)$ for each coordinate's LSTM, typically a multi-layer LSTM (e.g., two layers as used in the paper), consists of a tuple of (cell state, hidden state) pairs for each layer, i.e., $((c_{t,1}, h_{t,1}), (c_{t,2}, h_{t,2}))$ for a two-layer LSTM. These states allow the optimizer to accumulate information over the optimization trajectory, akin to momentum. The function $\mathcal{A}$ is then defined as

$$\mathcal{A}(\mathbf{w}_{opt}, x(t), t) = -\frac{1}{h} m_{\mathbf{w}_{opt}}(\nabla f(x(t)), \text{state}(t)). \tag{28}$$

Here, $\mathbf{w}_{opt}$ are the learnable weights of the shared LSTM optimizer.

**RNNprop Optimizer.** Building on similar principles, Lv et al. [29] proposed the RNNprop optimizer. This optimizer also typically uses a coordinate-wise multi-layer LSTM (e.g., two-layer) as its core recurrent unit. Before the gradient information $\nabla f(x(t))$ is fed to the RNN, it undergoes a preprocessing step, $\mathcal{P}$. This preprocessing involves calculating Adam-like statistics, such as estimates of the first and second moments of the gradients, $s(t) = (\hat{m}(t), \hat{v}(t))$, which are then used to normalize the current gradient and provide historical context. The preprocessed features, $\mathcal{P}(\nabla f(x(t)), s(t))$, along with the RNN's previous state, state$(t)$, are input to the RNN. Similar

to the LSTM-optimizer described above, state$(t)$ for each coordinate's RNN consists of the (cell state, hidden state) tuples for each of its layers. The output of the RNN is then passed through a scaled hyperbolic tangent function to produce the final update $U(t)$. Let this entire update-generating function be $U_{\mathbf{w}_{opt}}(\nabla f(x(t)), s(t), \text{state}(t))$. The corresponding $\mathcal{A}$ function is

$$\mathcal{A}(\mathbf{w}_{opt}, x(t), t) = -\frac{1}{h} U_{\mathbf{w}_{opt}}(\nabla f(x(t)), s(t), \text{state}(t)), \tag{29}$$

where $U_{\mathbf{w}_{opt}}(\cdot)$ can be more specifically written as $\alpha \tanh(\text{RNN}(\mathcal{P}(\nabla f(x(t)), s(t)), \text{state}(t); \mathbf{w}_{opt}))$. The parameters $\mathbf{w}_{opt}$ encompass those for the preprocessing module $\mathcal{P}$ and the RNN, and $\alpha$ is a scaling hyperparameter.

