# OpenReview forum: "Accelerating Optimization via Differentiable Stopping Time"
_NeurIPS.cc/2025/Conference — NeurIPS 2025 spotlight_

### Official Review · Reviewer_fnrf · 2025-07-02

**Clarity:** 2
**Significance:** 2
**Originality:** 3
**Rating:** 4
**Confidence:** 3

**Summary:**

This paper proposes a computationally tractable way of differentiating stopping times associated with a given stopping criterion for an optimization problem. Some potential applications for the proposed differentiable stopping time are given, followed by experiments to illustrate further potential benefits of adopting it.

**Questions:**

7.1: Could K_max in the objective (10) be replaced with N_J? Having K_max in the summation possibly accounts for performance after the stopping criterion is achieved, does that mean the criterion is not enforced to stop the optimization run? In [29] the authors report that optimizing only for the last loss f(x_{K_max}) gives better results. In that case, would N_J still lead to better performance or would it have the same effect as f(x_{K_max})?
7.2: On Fig 2.a, why do smaller epsilon lead to smaller relative error? There is less transient as the solution settle?
7.3: Is there a way to quantify the impact of the relative error on the quality of the learned optimizer? Is there a range of step-sizes in which the adaptive ODE solver approach is more advantageous? That would help to compare the two approaches more fairly and get a better sense of the potential significance of the paper.

**Ethical Concerns:**

["NO or VERY MINOR ethics concerns only"]

**Final Justification:**

This is a technically solid paper with an interesting core idea that could have broader impact. The authors have responded responsibly to feedback and committed to meaningful improvements. However, the evaluation limitations and questions about practical significance that motivated my original score remain largely unchanged. overall I am generally supportive of this work and happy to see it will get accepted.

**Limitations:**

yes

**Quality:**

3

**Strengths And Weaknesses:**

Strengths: The idea of differentiating the stopping time is nice and could be important in applications. There is a good discussion of such potential applications, some experiments to validate the idea and the theoretical results are reasonably well presented.
Weaknesses:
The presentation can be improved significantly in my view. Although the title quite accurately captures the spirit of the paper, reading the abstract and the introduction was more confusing than informative and I only started to get a better understanding of the topic and contributions of the paper when I read section 2 in the first pass. For example, the sentences in the abstract are only mildly connected. The use of the term "dual" is confusing, since it has different meanings in different contexts. Likewise some sentences like "An efficient algorithm is designed to backpropagate through it" could be clarified.
Also, I believe the experiments could be described a little better. I understand and agree that their point is to serve as proof-of-concept rather than provide definite evidence of achieving state-of-the-art, but some further details would be helpful to get a better sense of what is happening. In particular, the training procedure of the optimizers in the "Learning to Optimize" experiment could be explained in more detail.
For example, the description of a similar experiment in [29] is easier to understand.
Finally, I think the results from experiment "Validation of Theorems 2 and Proposition 1" could be discussed in more detail. For example, as the proposed Euler method becomes more computationally advantageous than the adaptive ODE solver method, the discrepancy between them also decreases. So, an assessment of the impact of the relative error on the quality of the learned optimizer would help to compare the two approaches more fairly.
Minor points:
- In the main text, define W2,inf norms or at least say they are Sobolev norms.
- Line 161. Clarify what the limit of |x_k -x(t_k)| is taken wrt. What is the connection for future improvement of (9)?
- Perhaps consider moving Algorithm 2 to the appendix? The two-column presentation is somewhat distracting.

---

> ### Author Rebuttal · Authors · 2025-07-30
>
> Thank you for the thoughtful and constructive review. The core idea being recognized as potentially important is greatly encouraging. The detailed feedback regarding presentation and clarity is much appreciated, and each point will be addressed carefully in the revision.
>
> ## Response to Identified Weaknesses
>
> **Abstract and Introduction:**
>
> The abstract and introduction will be rewritten to present a clearer narrative. The revised version will begin by highlighting a key practical challenge, namely minimizing time to solution in optimization, which remains difficult due to the non-differentiability of stopping times. This will be followed by a precise statement of the main contribution, a theoretically grounded differentiable stopping time. The goal is to clearly convey the motivation, core idea, and downstream impact from the outset.
>
> **Terminology Clarification:**
>
> The use of the term “dual” was intended to describe the contrast between minimizing loss within a fixed budget and minimizing the time needed to reach a target loss. To avoid ambiguity, this term will be replaced with a more precise description. Additionally, the phrase “backpropagate through it” will be clarified to indicate the use of the chain rule in computing gradients of a final objective with respect to algorithm hyperparameters, where the differentiable stopping time serves as an intermediate variable.
>
> **Training Details in the Learning to Optimize (L2O) Experiment:**
>
> An expanded explanation of the L2O training procedure will be included in the appendix, following the structure provided in [2]. This addition will include details on the meta objective optimization from Equation (10), the mini batching strategy across different optimization problems, the use of Truncated Backpropagation Through Time (BPTT), and the method for combining gradients from the standard loss and the stopping time penalty.
>
> **Discussion of the Validation Experiment:**
>
> The validation experiment demonstrates that the proposed method becomes more efficient relative to adaptive ODE solvers as the step size increases, at the cost of higher relative error. While the current study serves as a proof of concept for Theorem 2 and Proposition 1, the suggestion to assess the downstream impact of this relative error is insightful. The revised manuscript will include a discussion clarifying that the small relative error observed empirically suggests that the approximation is sufficiently accurate for training learned optimizers, while also emphasizing that a suitable step size is needed to be carefully selected to avoid instability when differentiating the stopping time.
>
> ## Response to Minor Comments
>
> - **$W^{2,\infty}$ Norms:** A clarification will be added to the main text indicating that these are Sobolev norms.
> - **Clarification of Line 161:** The statement regarding the limit of $\\|x_k - x(t_k)\\|$ refers to the result in [1], which shows convergence under certain conditions as $k \to \infty$, even with fixed step size $h$. This is relevant because it suggests the potential to strengthen the $O(h)$ error bound in Equation (9). If similar convergence guarantees could be extended to the present framework, it would indicate that the proposed gradient approximation remains accurate at larger step sizes. This connection will be made explicit in the revision.
> - **Algorithm 2 Placement:** To improve readability, Algorithm 2 will be moved to the appendix, avoiding formatting issues in the main two column layout.
>
> ## Response to Question 1: Objective Formulation in L2O
>
> **Replacing $K_{\max}$ with $N_J$:**
>
> Using $N_J$ in place of $K_{\max}$ within the summation in Equation (10) is indeed feasible. In fact, it is also possible to train the optimizer using only $N_J$ as the objective. This design has been explored in prior work such as ODE-based Learning to Optimize [1], where $N_J$ alone is used to guide learning. However, that method depends on solving an ODE system, which limits its scalability to large problems.
>
> In the present work, summing up to $K_{\max}$ was adopted to maintain consistency with standard settings in the Learning to Optimize literature, where training is typically conducted over a fixed number of iterations. This choice ensures that the training horizon and the amount of information available to the optimizer remain consistent across different methods. Importantly, the computation of $N_J$ and its gradient relies only on information from iterations $1$ through $K_{\max}$, so including the full trajectory up to $K_{\max}$ allows fair comparison between our approach and existing baselines. This design choice and its implications will be discussed in more detail in the revised manuscript.
>
> **Optimizing $f(x_{K_{\max}})$ vs. $N_J$:**
>
> Minimizing $f(x_{K_{\max}})$ provides a sparse but targeted training signal that focuses only on the final outcome. In contrast, minimizing $N_J$ offers trajectory level feedback that promotes efficiency throughout the optimization process. These objectives are complementary.
>
> While [2] reports that using only $f(x_{K_{\max}})$ can improve generalization, the relative advantage of $f(x_{K_{\max}})$ versus $N_J$ depends on the specific problem setting. When the goal is to minimize the number of iterations required to reach a given level of accuracy, $N_J$ provides a more informative and useful signal. This is often the case in classical numerical optimization problems. On the other hand, when the final performance after a fixed number of steps is the primary concern, as is often the case in machine learning applications, optimizing $f(x_{K_{\max}})$ may be more effective.
>
> These two training signals reflect different priorities and can be viewed as complementary. In fact, combining them, for example through $\min_\theta [f(x_{K_{\max}}) + \lambda N_J]$, can encourage both rapid convergence and low final error. The revised version of the paper will include a dedicated discussion to clarify these tradeoffs and to explain the suitability of different objectives under various practical scenarios.
>
> ## Response to Question 2: Effect of Epsilon on Relative Error
>
> Intuitively, smaller values of $\epsilon$ lead to longer optimization trajectories that settle closer to the optimum, where the dynamics are smoother and the discrete approximation becomes more accurate. However, a deeper explanation is supported by our results. As noted in the **Clarification of Line 161**, the theoretical analysis in [1] shows that $\\|x_k - x(t_k)\\|$ can decrease as $k$ increases, even under a fixed step size. This directly explains the trend observed in Figure 2: smaller $\epsilon$ leads to larger $k$, which in turn reduces the discrepancy between the discrete and continuous trajectories, and thus the relative error.
>
> This interpretation is also reinforced by Figure 1(a). As the optimization progresses, the distance between the discrete iterates and the continuous path visibly decreases. In particular, the discrete and continuous trajectories gradually align as they approach the stopping region, further supporting the claim that gradient approximation becomes more accurate near convergence.
>
> These observations validate the remark made after Theorem 2 and provide additional evidence that the proposed method remains accurate even at larger step sizes. The revised version of the paper will emphasize this connection more clearly.
>
> ## Response to Question 3: Relative Error and Step Size Tradeoff
>
> We appreciate the reviewer’s insightful question. To address it, we begin by clarifying that in many L2O methods such as RNNprop, the effective step size is internally controlled. Specifically, the learned optimizer outputs a vector \$\Delta \theta\_t = \alpha \tanh(x\_{\text{out}})\$, which inherently bounds the step size by a small constant \$\alpha\$ (set to \$0.1\$ in our experiments). As a result, even though our method introduces a step size parameter \$h\$ to align the discrete updates with the continuous-time framework, this \$h\$ is purely technical and does not alter the effective behavior of the optimizer. In fact, it cancels out analytically when back-substituted, as detailed in Appendix D.
>
> The reviewer's question about the relationship between gradient approximation error and optimizer quality is important. As mentioned in the response to Question 2, empirical observations indicate that the discrepancy between the discrete and continuous trajectories decreases as iterations progress, even for fixed \$h\$. This suggests that the error does not grow linearly with \$h\$ as predicted by Theorem 2, provided that the optimizer remains stable. Under this stability condition, a larger \$h\$ often leads to faster convergence and lower overall computational cost, without sacrificing solution quality.
>
> However, if the chosen \$h\$ is too large and causes the learned optimizer to diverge, then using an adaptive ODE solver becomes advantageous. Adaptive solvers can automatically reject unstable steps and adjust the step size to ensure convergence. This makes them a robust fallback when stability is not guaranteed. We will revise the paper to highlight this tradeoff and to make explicit the conditions under which each approach is preferable.
>
>
> ## References
>
> [1] Z. Xie, W. Yin, and Z. Wen, “ODE-based Learning to Optimize,” 2024, *arXiv:2406.02006*.
>
> [2] K. Lv, S. Jiang, and J. Li, “Learning gradient descent: Better generalization and longer horizons,” in *Proc. Int. Conf. Mach. Learn. (ICML)*, 2017, pp. 2247–2255.

---

> > ### Comment · Reviewer_Hhor · 2025-08-04
> >
> > Thanks for the clarification and detailed response. My concerns have been addressed. I will maintain the score.

---

### Official Review · Reviewer_Hhor · 2025-07-02

**Clarity:** 4
**Significance:** 4
**Originality:** 4
**Rating:** 5
**Confidence:** 2

**Summary:**

To accelerate the convergence of iterative optimization algorithms, this paper models the dynamic of continuous steps at optimum to variable by ODE and discretizes it to recover true iterations. Rigorous derivative construction and demonstration of the approximation between discretized and continuous steps are proposed. Moreover, an explicit temporal derivative calculation method is proposed to efficiently alleviate redundant derivative calculations. The proposed method can be straightforwardly applied to the learning to optimize  (L2O) framework and to accelerate existing numerical algorithms, whose effectiveness is evaluated on synthetic optimization problems.

**Questions:**

1. Why should L2O insert the extra minimization of number of step? Is it feasible to directly optimize meta-learning process with the proposed differentiable stopping time?
2. Is the improvement of proposed method limited? What is $h$ for L2O? As in the configuration in line 865, to align with proposed $h\mathcal{A}$ framework, the L2O model should shrink its original update by $\frac{1}{h}$.
3. In the demonstration for Theorem 2, the boundary is made up from second-order Taylor Expansion. Is the equality trivial?
4. Does Definition 1 require a continuous property of $J$ to hold the implicit function theory?

**Ethical Concerns:**

["NO or VERY MINOR ethics concerns only"]

**Final Justification:**

My final rating is 5. This is a well-written paper. The paper proposes a theoretically guaranteed plugin component to improve the convergence of existing L2O methods, which provides a new aspect of L2O.

**Quality:**

4

**Strengths And Weaknesses:**

Strengths:
1. This paper is well-written and well-organized. First, the motivation is clearly presented. Second, a big picture of the discretization of ODE for continuous step size is illustrated. Then, the discretization is achieved by Euler approximation. After that, an efficient algorithm is proposed to calculate the acceleration formulation. Further, the proposed method is utilized to accelerate learning to optimize (L2O) and Adam.
2. This paper proposes a simple but straightforward modeling method for the number of step minimization, where the derivative of step to variable is formulated to illustrate the dynamic of steps. The proposed definition of the optimal step where a small objective emerges connects the dynamic of the objective with variable to the dynamic of steps.
3. This paper proposes an efficient and explicit derivative calculation method for each step. The method takes an induction methodology and eliminates redundant computation in back-propagation through time.

Weaknesses:
1. Experiments are not enough. First, the optimization problems in the experiments are all synthetic. Since the proposed method is not limited to solving convex optimization problems, some real-world optimization problems (even non-convex ones) will be better. Second, the presented improvement in experimental results for L2O is not significant.
2. Figure 1 lacks explanation.

---

> ### Author Rebuttal · Authors · 2025-07-30
>
> ## Regarding the Experiments (Weaknesses 1)
>
> We appreciate the reviewer's suggestion to include more complex, real-world, and non-convex problems.
>
> - **Inclusion of Real-World Benchmarks:** We would like to gently clarify that not all experiments are purely synthetic. The "Online Learning Rate Adaptation" experiments were conducted on **smooth Support Vector Machine (SVM) problems using standard benchmark datasets from LIBSVM**. We will revise our manuscript to add more results of L2O on real-world dataset, such as datasets from LIBSVM. We will also include the results on the generalized Rastrigin function, a non-convex optimiation problem [1]:
> $$
> f_q(\boldsymbol{x})=\frac{1}{2}\left\\|\boldsymbol{A}_q \boldsymbol{x}-\boldsymbol{b}_q\right\\|_2^2-\alpha \boldsymbol{c}_q \cos (2 \pi \boldsymbol{x})+\alpha n,
> $$
> where $\boldsymbol{A}_q \in \mathbb{R}^{n \times n}, \boldsymbol{b}_q \in \mathbb{R}^{n \times 1}$ and $\boldsymbol{c}_q \in \mathbb{R}^{n \times 1}$ are parameters whose elements are sampled i.i.d. from $\mathcal{N}(0,1)$.
> - **Significance of the L2O Results:** We acknowledge the reviewer’s concern on the significance of improvement, but emphasize that the key advance lies in the *qualitative change* in optimization behavior. Unlike the baseline (L2O-RNNprop), which exhibits strong initial progress but quickly stagnates due to a short-sighted policy, our method (L2O-RNNprop-Time) avoids stagnation by learning to prioritize rapid convergence to a target precision through a differentiable stopping time objective. This shift encourages the optimizer to exploit favorable local geometries and transition effectively into efficient local convergence. It leads to sustained progress and final objective values nearly an order of magnitude lower. Crucially, this training strategy enhances generalization, as seen in Figure 3b, where our method scales to problems four times larger than training, outperforming the baseline significantly. By combining intermediate loss signals with differentiable stopping time, our approach learns more robust and adaptable update rules that generalize across horizons and scales.
>
> ## Regarding the Explanation of Figure 1 (Weakness 2)
>
> We agree that the explanation of Figure 1 in the main text could be more explicit, and we appreciate the opportunity to clarify its purpose. The figure is designed to provide intuitive visual support for the core concepts introduced in Section 2.
>
> - **Figure 1a (Trajectory vs. discrete path):** This panel illustrates how the algorithm’s hyperparameter, $\theta$, directly influences the optimization path. The solid lines represent the idealized continuous trajectories derived from the corresponding ODE, while the dotted lines with markers depict the actual discrete updates of the algorithm. The blue ellipses indicate the level sets of the stopping criterion (e.g., $\\|\nabla f(x)\\|= c$). As shown, varying $\theta$ alters the trajectory from 0.5 (red) to 2.5 (green), leading to different intersection points with the stopping condition, both in location and timing. This highlights the sensitivity of the solution path to hyperparameter choices.
> - **Figure 1b (Stopping time vs. parameters):** This plot captures a central idea of our work: the stopping time (denoted $T_J$ for the continuous case and $N_J$ for the discrete) is a *function* of the hyperparameter $\theta$. Each curve corresponds to a different stopping tolerance $\epsilon$. The close alignment between the discrete stopping times $N_J$ (shown with transparent markers) and their continuous approximations $T_J$ (solid lines) provides visual evidence for the validity of our ODE-based approximation. Importantly, the smoothness and differentiability of these curves with respect to $\theta$ are key properties that our method exploits to enable efficient hyperparameter optimization.
>
> We will revise the manuscript to include a more detailed walkthrough of Figure 1 within Section 2, ensuring stronger integration between the visual illustrations and the theoretical framework. This enhancement will help readers better connect the geometric intuition with the formal definitions.
>
> ## Regarding the Question 1
>
> This approach is feasible and is verified in [2], which only uses the differentiable stopping time. However, this method still relies on the ODE and thus requires heavy computational overhead, hindering its application to large-scale problems. In this paper, we find it is also feasible to use only the stopping time. However, baseline L2O methods utilize information gathered over a fixed number of steps for training—a standard paradigm.  To conduct a fair comparison with other L2O methods, we insert the extra minimization of number of step. By inserting the stopping time, we do not introduce more information than the baseline methods, especially since all information needed to differentiate the stopping time is already included in the $K_{\max}$ steps. This confirms the effectiveness of our method.
>
> ## Regarding the Question 2
>
> For the specific L2O-RNNprop and L2O-RNNprop-Time methods, we strictly follow the setting in [3, Eq. 7], where the RNN outputs a single vector $x_{\text{out}}$, and the update increment is:
> $$
> \Delta \theta_t=\alpha \tanh \left({x}_{\mathrm{out}}\right)
> $$
>
> This formula acts as a form of gradient clipping, bounding all effective step sizes by the preset parameter $\alpha$. In our experiments, we set $\alpha=0.1$.
>
> Therefore, the effective step size is inherently capped at 0.1. This is a common characteristic in L2O methods, many of which have a relatively small, built-in effective step size. As a result, our proposed method can be applied to them directly without modification while maintaining stability, ensuring performance is not hindered.
>
> A general principle for increasing efficiency is to select a step size $h$ that is as large as possible, while still ensuring the stability of both the algorithm and the backpropagation for the differentiable stopping time. This principle aligns with the conventional wisdom in optimization: a larger step size often leads to faster convergence and, consequently, less computation.
>
> For the L2O experiments, the step size $h$ is a conceptual element of the framework, used to connect discrete-time learned optimizers to our continuous-time theory. As detailed in Appendix D, learned optimizers typically output a direct update $U_k$ such that $x_{k+1} = x_k + U_k$. To align this with our framework's formulation, $x_{k+1} = x_k - h\mathcal{A}(\theta, x_k, t_k)$, we define $\mathcal{A}(\theta,x_k,t_k) = -U_k/h$.
>
> When this definition of $\mathcal{A}$ is substituted back into the discretization formula, the $h$ in the numerator and denominator cancels out, recovering the original update:
> $$
> x_{k+1} = x_k - h(-U_k/h) = x_k + U_k
> $$
> This formulation allows the theory of differentiable stopping time to be applied without altering the behavior of existing L2O models, even though the gradient calculation for the stopping time depends on this conceptual $h$.
>
> ## Regarding the Question 3
>
> The proof technique for Theorem 2 is an application of Taylor's theorem for estimating the local error of a forward Euler scheme. However, the **key contribution lies in the implication** of this result within our framework. The $O(h)$ error bound in our theorem relies on the conventional assumption that the global error between the discrete and continuous states, $\\|x_N - x(T)\\|$, is also $O(h)$.
>
> This is where we connect to a **highly non-trivial result** from the literature. As we cite, [2] proves that for certain forms of the optimization algorithm $\mathcal{A}$, the global error $\\|x_k - x(t_k)\\|$ can converge to zero even with a constant, non-vanishing step size $h$. This phenomenon is also illustrated in our Figure 1, where the discrete path (dotted line) progressively aligns with the continuous trajectory.
>
> Our framework considers a more general form of $\mathcal{A}$ than that in [2]. While a formal proof of this stronger convergence for our general case is beyond the current scope, our strong numerical results suggest these favorable properties may carry over. Investigating the precise conditions for this convergence in our broader framework is an important direction for future work. We will revise the remark follows Theorem 2 to make this distinction clearer.
>
> ## Regarding the Question 4
> Yes, the continuous differentiability of the function $J(x)$ is a required condition. While Definition 1 itself only defines the stopping time, the subsequent Theorem 1, which establishes the differentiability of the stopping time, relies on this property. To exclude the queer situation where $J$ is a discontinuous function and a gap in the stopping time may occur, we will revise our manuscript to explicitly state the continuity of $J$ with respect to $x$ in Definition 1.
>
> ## Reference
>
> [1] T. Chen et al., "Learning to optimize: A primer and a benchmark," *J. Mach. Learn. Res.*, vol. 23, no. 189, pp. 1–59, 2022.
>
> [2] Z. Xie, W. Yin, and Z. Wen, "ODE-based Learning to Optimize," 2024, *arXiv:2406.02006*.
>
> [3] K. Lv, S. Jiang, and J. Li, "Learning gradient descent: Better generalization and longer horizons," in *Proc. Int. Conf. Mach. Learn. (ICML)*, 2017, pp. 2247–2255.

---

### Official Review · Reviewer_zF24 · 2025-07-03

**Clarity:** 4
**Significance:** 3
**Originality:** 3
**Rating:** 5
**Confidence:** 4

**Summary:**

This paper introduces a novel training paradigm for learning-based optimization solvers. Rather than differentiating the objective function at a fixed iteration, the authors propose to differentiate with respect to the stopping time required to reach a given accuracy. The proposed method is mathematically well-grounded and is evaluated on standard benchmark problems, where it demonstrates certain advantages over existing approaches.

**Questions:**

See "Strengths And Weaknesses"

**Ethical Concerns:**

["NO or VERY MINOR ethics concerns only"]

**Final Justification:**

The authors have addressed my concerns, so I will keep my positive feedback

**Limitations:**

See "Strengths And Weaknesses"

**Quality:**

3

**Strengths And Weaknesses:**

The paper is clearly written, and the theoretical derivations are rigorous and convincing. My questions primarily concern the experimental results:

(L2O-DM Performance). All baseline L2O methods—including L2O-DM—were trained using 500 iterations, correct? If so, it is puzzling that in Figure 3, L2O-DM achieves its best performance at iteration 10 rather than near the training horizon of 500. Does this pattern also appear on the training set? Has such behavior been reported in prior literature? (To my knowledge, this is not a commonly observed phenomenon.)

(L2O-RNNprop vs. L2O-RNNprop-Time). Still referring to Figure 3, the performance gap between L2O-RNNprop and L2O-RNNprop-Time appears to peak around accuracy $10^{-5}$, which is consistent with the training tolerance of L2O-RNNprop-Time. Furthermore, the performance gap narrows around iteration 500, matching its training horizon. If this interpretation is correct, it would be helpful to include a brief discussion of these observations in the paper.

(Generalization Beyond Training Scope). It is interesting to note that while L2O-RNNprop-Time is trained with a target tolerance of $10^{-5}$, it generalizes well to higher-accuracy regimes. In contrast, L2O-RNNprop seems unable to improve further beyond the accuracy it achieves at 500 iterations, even when allowed more iterations. Could you elaborate on why the proposed training method leads to better generalization in this regard?

---

> ### Author Rebuttal · Authors · 2025-07-30
>
> We sincerely appreciate the time and effort the reviewers have dedicated to evaluating our paper. We are encouraged by the positive feedback and grateful for the thoughtful and constructive suggestions.
>
> **L2O-DM Performance**
> The observation is accurate. This issue is reported in several studies, including Figure 6 and Figure 8 of [1], where L2O-DM consistently fails to decrease the loss function. Similar phenomena are also observed in Figure 2 and Figure 5 of [2]. This behavior also appears in the training set, where performance is similar to that in testing: the loss decreases during the first 10 steps and then begins to increase. These works confirm the weak generalization of the L2O-DM method, which stems from the intrinsic limitation of model-free L2O approaches. Their inability to capture gradient information, causing them to rely heavily on the alignment between gradient directions and the expressive space of the model-free optimizer. When the gradient is small, the noise inherent in model-free methods may overwhelm the gradient signal, leading to divergence.
>
> There is a minor clarification (which does not affect the argument): in Line 246, we train the L2O optimizers for 500 steps, and during training, at each step, the L2O optimizers are allowed to run for $K_{\text{max}} = 100$ steps to obtain the training loss. We will emphasize this point in the revised manuscript.
>
> **L2O-RNNprop vs. L2O-RNNprop-Time**
> The interpretation is largely accurate. The peak in the performance gap indeed aligns with the accuracy threshold used to determine the stopping criterion during training. However, we would like to clarify that the training horizon of L2O-RNNprop-Time is 100 iterations. The observed narrowing of the performance gap around iteration 500 is primarily due to the loss approaching a plateau, where gradients become very small and further improvement is limited. We will include a brief discussion in the revised version to clarify this point and aid interpretation.
>
> **Generalization Beyond Training Scope**
> The improved generalization of L2O-RNNprop-Time can be attributed to its training objective, which shifts the optimization focus from minimizing the loss over a fixed number of steps to minimizing a differentiable stopping time under a specified target precision. This design encourages the optimizer to prioritize rapid convergence to a desired accuracy, rather than just short-term loss reduction, thereby fostering a more robust update policy that generalizes better to longer horizons and higher-accuracy regimes.
>
> Previous studies such as Lv et al. [2] show that focusing exclusively on the final‑step loss $f(x_{K_{\max}})$ can sometimes improve generalization by prioritizing the accuracy of the ultimate iterate. Our approach differs in that it retains the intermediate loss signals, while still incorporating an explicit focus on the final target accuracy through the stopping time penalty. This hybrid objective provides a richer training signal, helping the learned optimizer balance early progress with long term convergence, and often better exploit local convergence behaviors once a target precision is reached. We believe this complementary use of trajectory information and precision-specific emphasis contributes to the observed generalization gains.
>
> **Reference**
> [1] T. Chen et al., "Learning to optimize: A primer and a benchmark," *J. Mach. Learn. Res.*, vol. 23, no. 189, pp. 1–59, 2022.
> [2] K. Lv, S. Jiang, and J. Li, "Learning gradient descent: Better generalization and longer horizons," in *Proc. Int. Conf. Mach. Learn. (ICML)*, 2017, pp. 2247–2255.

---

> > ### Comment · Reviewer_zF24 · 2025-08-06
> >
> > Thanks for your detailed response. My questions and concerns have been addressed, so I will keep my positive score.

---

### Decision · Program_Chairs · 2025-09-17

**Decision:**

Accept (spotlight)

**Comment:**

To accelerate the convergence of iterative optimization algorithms, this paper models the optimization dynamic of continuous steps by ODE, computes the derivatives of stopping time with respect to learnable parameters, and proposes approximation algorithms for the practical discrete use case. The paper provides guarantee for the discrete approximation, and the proposed method can be straightforwardly applied to the learning to optimize (L2O) framework and to accelerate existing numerical algorithms. The reviewers think the method is original, and has good theoretical support and practical applications.